# Multi-Objective Multi-Fidelity Bayesian Optimization with Causal Priors

## Abstract

Multi-fidelity Bayesian optimization (MFBO) accelerates the search for the global optimum of black-box functions by integrating inexpensive, low-fidelity approximations. The central task of an MFBO policy is to balance the cost-efficiency of low-fidelity proxies against their reduced accuracy to ensure effective progression toward the high-fidelity optimum. Existing MFBO methods primarily capture associational dependencies between inputs, fidelities, and objectives, rather than causal mechanisms, and can perform poorly when lower-fidelity proxies are poorly aligned with the target fidelity. We propose RESCUE (REducing Sampling cost with Causal Understanding and Estimation), a multi-objective MFBO method that incorporates causal calculus to systematically address this challenge. RESCUE learns a structural causal model capturing causal relationships between inputs, fidelities, and objectives, and uses it to construct a probabilistic multi-fidelity (MF) surrogate that encodes intervention effects. Exploiting the causal structure, we introduce a causal hypervolume knowledge-gradient acquisition strategy to select input–fidelity pairs that balance expected multi-objective improvement and cost. We show that RESCUE improves sample efficiency over state-of-the-art MF optimization methods on synthetic and real-world problems in robotics, machine learning (AutoML), and healthcare.

## 1. Introduction

Optimizing black-box functions that lack analytical forms and derivative information is a ubiquitous problem, especially when they are expensive to evaluate. Multi-Objective Bayesian Optimization (MOBO) addresses this problem in a sample-efficient manner (Hernández-Lobato et al., 2016)

[1]Anonymous Institution, Anonymous City, Anonymous Region, Anonymous Country. Correspondence to: Anonymous Author <anon.email@domain.com>.

Preliminary work. Under review by the International Conference on Machine Learning (ICML). Do not distribute.

by iteratively maximizing an acquisition function computed from a probabilistic surrogate model (typically a Gaussian Process (GP)) to select candidate inputs for intervention. The selected inputs are evaluated on the true objective functions, and the surrogate is updated to improve objective estimation. However, executing interventions on the true objective functions is costly, as each intervention can involve extensive computation or expensive real-world experiments. In practice, multiple Information Sources (IS) with varying numerical resolutions are often available to approximate the objective functions. Interventions at low fidelities typically incur lower cost but provide less accurate approximations than interventions at higher fidelities. For example, tuning parameters of a mobile robot can be costly in real-world scenarios (e.g., longer evaluation time, safety concerns, hardware degradation, operational constraints) but can be accelerated using lower-fidelity simulations such as *Gazebo* (Hossen et al., 2025). A key challenge is optimizing the trade-off between the cost and accuracy of MF interventions to enable informed decision-making. This challenge intensifies when auxiliary IS poorly approximate the target fidelity. In such cases, single-fidelity BO (SFBO)—i.e., BO using only the primary IS—can outperform MF methods under a fixed budget (Mikkola et al., 2023).

In this paper, we present RESCUE, a multi-objective multi-fidelity optimization method that integrates causal inference within the MOBO framework. RESCUE builds upon the literature on constrained causal BO (Aglietti et al., 2023), causal knowledge transfer for BO (Hossen et al., 2025), and MF optimization (Mikkola et al., 2023; Poloczek et al., 2017). Specifically, RESCUE captures both the causal relationships among inputs and the causality between inputs and outputs by learning a causal model from observational data. We use this causal model to construct an MF multi-output GP surrogate with a causal prior, henceforth the MF Causal GP (MF-CGP). The MF-CGP estimates the objectives while also capturing cross-fidelity correlations. RESCUE strategically performs causal inference and leverages the MF-CGP to select an input-fidelity pair by maximizing the MF Causal Hypervolume Knowledge Gradient (C-HVKG) acquisition function, which is an extension of HVKG (Daulton et al., 2023). The causal model and the MF-CGP are actively updated using new observations to refine the causal structure and improve output estimation.

Our contributions are as follows:

- We develop a multi-objective multi-fidelity optimization method that integrates causal calculus into MF-MOBO framework to estimate intervention effects.

- We propose the multi-fidelity Causal Hypervolume Knowledge Gradient (C-HVKG), an acquisition strategy that exploits the causal structure to drive exploration by selecting an optimal input-fidelity pair while limiting infeasible interventions.

- Theoretically, we show RESCUE's performance can be bounded to its single-fidelity MOBO counterpart.

- We demonstrate substantial gains in optimization performance of RESCUE over state-of-the-art MF-MOBO and non-MF MOBO on a variety of synthetic and real-world MF problems in robotics, ML, and healthcare.

## 2. Related Works

Integrating multiple IS into BO to reduce optimization cost has been studied via hierarchical MFBO methods when sources can be ranked by fidelity (Li et al., 2020; Takeno et al., 2020; Moss et al., 2021; Tighineanu et al., 2022) and via non-hierarchical methods otherwise (Lam et al., 2015; Poloczek et al., 2017; Wu et al., 2023). Although hierarchical methods may suffer from error propagation from lower-fidelities (Wu et al., 2023), we do not claim that non-hierarchical methods are inherently superior or inferior; rather, the choice is application specific (Zhou et al., 2023). Our MF-CGP jointly learns cross-fidelity correlations, while the causal model lets experts encode domain knowledge.

Multi-fidelity extensions of BO have been widely studied for multi-objective (Irshad et al., 2024; Daulton et al., 2023; Belakaria et al., 2020; He et al., 2022) and single-objective (Mikkola et al., 2023; Li et al., 2020; Takeno et al., 2020; Poloczek et al., 2017) optimization. We refer the interested reader to Daulton et al. (2023, Section 3) and Belakaria et al. (2020, Section 3) for details. When IS poorly approximate the target fidelity, MF optimization can incur a higher cumulative evaluation cost than SFBO. rMFBO (Mikkola et al., 2023) mitigates this with a theoretical guarantee that, even with unreliable sources, its performance is bounded by SFBO, but it applies only to the single-objective setting. Despite their success, none of these methods captures the causal structure existing among inputs, fidelities, and objectives. These methods rely on ML models and sampling heuristics to predict objectives across fidelities, which may capture spurious correlations when distribution shifts occur across fidelities (Iqbal et al., 2023).

Although there is extensive literature on MFBO and causal BO (Aglietti et al., 2020; 2023), the literature on integrating causal inference in MF setting is limited. Recent works have focused on learning a causal model from the source (e.g., a robot in simulation), transferring it to the target (e.g., a real robot), and using it to identify causally relevant inputs to reduce the MOBO search space (Hossen et al., 2025). However, direct transfer of the causal model can induce bias. While our contributions build upon previous work on causal BO and causal knowledge transfer for accelerating optimization, none of these works consider the MF setting.

## 3. Preliminaries

Random variables and observations denoted by upper- and lower-case letters, respectively, and vectors appear in bold.

### 3.1. Problem Setup

Consider a highly configurable system with $X_i$ indicating the $i^{\text{th}}$ configuration option, which takes values $x_i \in \text{Dom}(X_i)$ from a finite domain. The configuration space is the Cartesian product of all parameter domains, $\mathcal{X} = \times_{i \in [d]} X_i$, where $d$ is the number of configuration options. A configuration $\mathbf{x} \in \mathcal{X}$ is instantiated by setting a specific value for each option within its domain, $\mathbf{x} = \langle X_1 = x_1, X_2 = x_2, \ldots, X_d = x_d \rangle$. Let $\mathcal{S}$ denote the fidelity space, and let $s_\diamond \in \mathcal{S}$ denote the target fidelity (e.g., a real system). For any fidelity $s \in \mathcal{S}$, we represent the performance objective as a black-box mapping from the joint configuration-fidelity space to $M$ real-valued objectives, $\boldsymbol{f} : \mathcal{X} \times \mathcal{S} \to \mathbb{R}^M$, where $\boldsymbol{f} = \{f_m\}_{m=1}^{M \geq 2}$. In practice, we learn $\boldsymbol{f}$ by sampling from the configuration-fidelity space and collecting observations $\mathbf{y} = \boldsymbol{f}(\mathbf{x}, s) + \boldsymbol{\epsilon}$, where $\boldsymbol{\epsilon} \sim \mathcal{N}(0, \sigma^2)$. Evaluations at fidelities $s \neq s_\diamond$ are regarded as auxiliary sources that approximate the target ($s_\diamond$) with variable bias $\boldsymbol{\delta}(\mathbf{x}, s) = \boldsymbol{f}(\mathbf{x}, s) - \boldsymbol{f}(\mathbf{x}, s_\diamond)$. We consider $\boldsymbol{f}(\mathbf{x}, s_\diamond)$ subject to $Q$ inequality constraints $\{h_q(\mathbf{x}, s)\}_{q=1}^Q$, which are observable across all fidelities. Let $c(\mathbf{x}, s)$ be the evaluation cost, and assume $c(\mathbf{x}, s) < c(\mathbf{x}, s_\diamond)$ for $s \neq s_\diamond$, where $s$ could correspond to simulation tools or simplified settings. Furthermore, $\boldsymbol{f}(\mathbf{x}, s_\diamond)$ can be observed directly without bias, although noise may influence it. Our goal is to find a configuration $\mathbf{x}^*$ *at the target fidelity $s_\diamond$* that yields Pareto-optimal performance while satisfying the constraints under a budget $\Lambda \leq C$, where $C := \sum_{t=1}^T c(\mathbf{x}_t, s_t)$ with $T$ denoting total interventions. Formally, we define[1]:

$$\mathbf{x}^* = \underset{\mathbf{x} \in \mathcal{X}}{\arg\min} \, \boldsymbol{f}(\mathbf{x}, s_\diamond) \text{ s.t. } h_q(\mathbf{x}, s_\diamond) \geq \gamma_q, \, \forall q \in [Q], \quad (1)$$

where $\gamma_q$ is the constraint threshold.

---

[1]Depending on application, Eq. (1) may be adjusted when objectives are maximized and/or some or all constraints must be below the threshold.

## 3.2. Background

Let $\mathcal{G} = (\mathbf{V}, \mathbf{E})$ denote a directed acyclic graph (DAG), where $\mathbf{V}$ is the set of observable variables and $\mathbf{E}$ is the set of directed edges capturing causal dependencies. A causal performance model (CPM) is a specialized structural causal model (Pearl, 2009; Peters et al., 2017) associated with $\mathcal{G}$ to formally model causal dependencies. The CPM is defined as $\mathcal{C} = \langle \mathbf{U}, \mathbf{V}, \mathbf{F} \rangle$, where $\mathbf{U}$ is a set of mutually independent exogenous variables determined by factors outside the model, and $\mathbf{F}$ is a set of deterministic functions such that $V_i = f_i(\mathrm{Pa}(V_i), U_i)$ for each $V_i \in \mathbf{V}$, with $\mathrm{Pa}(V_i) \subseteq \mathbf{V} \setminus \{V_i\}$ denoting the set of parents of $V_i$ and $U_i \in \mathbf{U}$. We draw a directed edge from $V_i$ to $V_j$ if $V_i \in \mathrm{Pa}(V_j)$. We assume *causal sufficiency*, so there are no unmeasured common causes of pairs of observed variables. We partition $\mathbf{V}$ into three disjoint subsets: configuration options $\mathbf{X}$, key performance indicators $\mathbf{Z}$ (non-manipulable variables), and performance objectives $\mathbf{Y}$. The fidelity $s \in \mathcal{S}$ is also an observed variable within $\mathbf{V}$, influencing the key performance indicators and performance objectives. While $\mathcal{G}$ captures the abstract causal structure, the CPM augments $\mathcal{G}$ with functional and probabilistic components, enabling causal reasoning and interventional analysis. We assume the system can be described by an CPM.

Using the rules of *do*-calculus (Pearl, 2009), a causal intervention (CI) (Peters et al., 2017) modifies $\mathcal{C}$ by replacing $f_{\mathbf{X}}(\mathrm{Pa}(\mathbf{X}), U_{\mathbf{X}})$ with $\mathbf{x}$, denoted by $\mathrm{do}(\mathbf{X} = \mathbf{x})$ which represents an intervention on $\mathbf{X}$ by setting its value to $\mathbf{x}$. Exploiting the rules of *do*-calculus, we can compute the interventional distribution $p(\mathbf{Y} | \mathrm{do}(\mathbf{X} = \mathbf{x}), s)$ at fidelity $s$. The expected value of $\mathbf{Y}$ under this interventional distribution is given by a causal function $\mathbb{E}_{p(\mathbf{Y} | \mathrm{do}(\mathbf{X}=\mathbf{x}), s)}[\mathbf{Y}]$. Given $\mathcal{G}$, we can write the observational distribution as $p(\mathbf{Y} | \mathbf{X} = \mathbf{x}, s)$ using the Causal Markov Condition (Pearl, 2009). Evaluating $p(\mathbf{Y} | \mathbf{X} = \mathbf{x}, s)$ requires *observing* while $p(\mathbf{Y} | \mathrm{do}(\mathbf{X} = \mathbf{x}), s)$ requires *interventions* at fidelity $s$. Computing $p(\mathbf{Y} | \mathrm{do}(\mathbf{X} = \mathbf{x}), s)$ often involves evaluating intractable integrals which can be approximated using observational data[2] $\widehat{\mathcal{D}}$ to get a Monte Carlo (MC) estimate $\widehat{p}(\mathbf{Y} | \mathrm{do}(\mathbf{X} = \mathbf{x}), s) \approx p(\mathbf{Y} | \mathrm{do}(\mathbf{X} = \mathbf{x}), s)$ under the appropriate identifiability conditions (Galles & Pearl, 1995). For notational simplicity, we denote $\boldsymbol{f}_{\mathrm{do}}(\mathbf{x}, s) = \mathbb{E}_{\widehat{p}(\mathbf{Y} | \mathrm{do}(\mathbf{X}=\mathbf{x}), s)}[\mathbf{Y}]$.

In this paper, we learn $\mathcal{G}$ from $\widehat{\mathcal{D}}$ using existing causal discovery methods—the PC algorithm (Spirtes et al., 2001) and DirectLiNGAM (Shimizu et al., 2011)—and integrate causal discovery into our framework. We enforce domain-specific structural constraints, such as disallowing directed edges from $\mathbf{Y}$ to $\mathbf{X}$, to facilitate learning $\mathcal{G}$ with limited data; see Hossen et al. (2023, Section 3) for details.

---

[2]We assume access to either observational data $\widehat{\mathcal{D}}$ (e.g., system logs, which are readily available in most systems) or a given DAG.

## 4. Methodology

In this section, we introduce a probabilistic surrogate model by exploiting causal calculus. We then design an acquisition strategy that uses causal information to balance the trade-off between cost and information in multi-fidelity query selection. Finally, we present the RESCUE algorithm to find optimal configurations under a finite evaluation budget.

### 4.1. Multi-output Multi-fidelity Causal GP

**Likelihood**   Let $\mathcal{D} = \{(\mathbf{x}_i, s_i, \mathbf{y}_i)\}_{i=1}^N$ be a finite collection of $N$ interventional samples. We assume that each $\mathbf{y}_i$ is a noisy observation of $\boldsymbol{f}(\mathbf{x}_i, s_i)$, i.e., $\mathbf{y}_i = \boldsymbol{f}(\mathbf{x}_i, s_i) + \boldsymbol{\epsilon}_i$, where $\boldsymbol{\epsilon}_i \sim \mathcal{N}(0, \sigma^2 \mathbf{I})$ and $\{\boldsymbol{\epsilon}_i\}_{i=1}^N$ are independent[3]. The feasibility indicators $\boldsymbol{g}(\mathbf{x}, s) = \{\mathbb{I}(h_q(\mathbf{x}, s) \geq \gamma_q)\}_{q=1}^Q$ are observed at each $(\mathbf{x}_i, s_i)$ and modeled as additional outputs using the same probabilistic model described below. For notational simplicity, we omit $\boldsymbol{g}$ from $\mathcal{D}$ and the subsequent formulation. The likelihood of $\mathcal{D}$ is $p(\mathcal{D} \mid \boldsymbol{f}, \sigma^2) = \prod_{i=1}^N \mathcal{N}(\mathbf{y}_i ; \boldsymbol{f}(\mathbf{x}_i, s_i), \sigma^2 \mathbf{I})$.

**Probabilistic model**   We place a causal prior (Aglietti et al., 2020) on the objective function $\boldsymbol{f}(\mathbf{x}, s)$. The causal prior is computed to estimate the causal effect of a configuration option on the objective. We construct the prior mean and variance of a GP using the observational data $\widehat{\mathcal{D}}$ and the corresponding CPM $\mathcal{C}$. For notational brevity[4], we define $\boldsymbol{x} := (\mathbf{x}, s)$ and $\boldsymbol{x}' := (\mathbf{x}', s')$. The MF-CGP is defined as

$$\boldsymbol{f}(\boldsymbol{x}) \sim \mathrm{GP}(\boldsymbol{\mu}(\boldsymbol{x}), k_{m,m'}(\boldsymbol{x}, \boldsymbol{x}')),$$
$$\boldsymbol{\mu}(\boldsymbol{x}) = \boldsymbol{f}_{\mathrm{do}}(\boldsymbol{x}),$$
$$k_{m,m'}(\boldsymbol{x}, \boldsymbol{x}') = (k_{\mathrm{in}}(\mathbf{x}, \mathbf{x}') k_{\mathrm{fid}}(s, s') + \widehat{\boldsymbol{\sigma}}(\boldsymbol{x}) \widehat{\boldsymbol{\sigma}}(\boldsymbol{x}'))$$
$$\otimes k_{\mathrm{obj}}(m, m'),$$

where $\widehat{\boldsymbol{\sigma}}(\boldsymbol{x}) = (\mathbb{V}_{\widehat{p}(\mathbf{Y} | \mathrm{do}(\mathbf{X}=\mathbf{x}), s)}[\mathbf{Y}])^{1/2}$ with $\mathbb{V}$ representing variance estimated from $\widehat{\mathcal{D}}$. The kernel $k_{m,m'}(\boldsymbol{x}, \boldsymbol{x}') = \mathrm{Cov}((\boldsymbol{x}, m), (\boldsymbol{x}', m'))$ captures spatial similarity and inter-objective relationships. Here, $k_{\mathrm{in}}(\cdot, \cdot)$ captures covariance between configurations, $k_{\mathrm{fid}}(\cdot, \cdot)$ captures similarity across fidelities, and $k_{\mathrm{obj}}(\cdot, \cdot)$ captures inter-objective covariance. For both $k_{in}(\cdot, \cdot)$ and $k_{\mathrm{fid}}(\cdot, \cdot)$ we use the radial basis function (RBF) kernel defined as $\exp(-\frac{||\boldsymbol{x} - \boldsymbol{x}'||^2}{2l^2})$.

Given this GP prior, the posterior distribution over $\boldsymbol{f}(\boldsymbol{x})$ can be derived analytically via standard GP updates. For notational brevity, we write $\boldsymbol{x}_i := (\mathbf{x}_{:,i}, s_i)$ and $\boldsymbol{x}_{1:N} := (\boldsymbol{x}_1, \ldots, \boldsymbol{x}_N)$. The posterior distribution is given as, $p(\boldsymbol{f}(\boldsymbol{x}) | \mathcal{D}) = \mathcal{N}(\boldsymbol{\mu}(\boldsymbol{x} | \mathcal{D}), \mathbf{K}(\boldsymbol{x}, \boldsymbol{x}' | \mathcal{D}))$, where $\boldsymbol{\mu}(\boldsymbol{x} | \mathcal{D}) = \boldsymbol{\mu}(\boldsymbol{x}) + \mathbf{k}(\boldsymbol{x}, \boldsymbol{x}_{1:N})^\top (\boldsymbol{\Sigma} + \sigma^2 \boldsymbol{I})^{-1}(\mathbf{y} - \boldsymbol{\mu}(\boldsymbol{x}_{1:N}))$ and $\mathbf{K}(\boldsymbol{x}, \boldsymbol{x}' | \mathcal{D}) = k(\boldsymbol{x}, \boldsymbol{x}') - \mathbf{k}(\boldsymbol{x}, \boldsymbol{x}_{1:N})^\top (\boldsymbol{\Sigma} + \sigma^2 \boldsymbol{I})^{-1} \mathbf{k}(\boldsymbol{x}', \boldsymbol{x}_{1:N})$ with

---

[3]To avoid clutter, $\boldsymbol{\epsilon}_i$ is omitted in the remainder of the paper.
[4]In this paper, $(\mathbf{x}, s)$ and $\boldsymbol{x}$ are used interchangeably.

**Algorithm 1** RESCUE Algorithm

**Require:** $\widehat{\mathcal{D}}$, $\mathcal{X}$, budget $\Lambda$, update cycle $N_l$
**Ensure:** Optimal configuration $\mathbf{x}^*$
1: Initialize $\mathcal{D}_0 = \{(\mathbf{x}_i, s_i, \mathbf{y}_i)\}_{i=1}^N$, $t = 0$, and $C_0 = \sum_{i=1}^N c(\mathbf{x}_i, s_i)$
2: Construct a causal performance model (CPM): $\mathcal{C}(\mathcal{G}|\widehat{\mathcal{D}})$
3: **while** $C_t \le \Lambda$ **do**
4:    $t \leftarrow t + 1$
5:    **if** $t \mod N_l == 0$ **then**
6:       $\mathcal{C}_t(\mathcal{G}_t|\widehat{\mathcal{D}} \cup \mathcal{D}_{0:t-1}) \leftarrow$ Update the CPM
7:    **end if**
8:    $\mathcal{M}(\mathbf{x}, s \mid \mathcal{D}_{0:t-1}, \mathcal{C}_t) \leftarrow$ Train a MF-CGP model
9:    Find next $(\mathbf{x}_t, s_t)$ by optimizing Eq. (2)
10:   Obtain $\mathbf{y}_t$ by intervening $\mathbf{x}_t$ at $s_t$
11:   Augment $\mathcal{D}_{0:t} \leftarrow \mathcal{D}_{0:t-1} \cup \{(\mathbf{x}_t, s_t, \mathbf{y}_t)\}$
12:   Update total cost $C_t \leftarrow C_{t-1} + c(\mathbf{x}_t, s_t)$
13: **end while**
14: Obtain Pareto front using NSGA-II (Deb et al., 2002) on the posterior mean at the target fidelity $\mathbf{x}^* = \text{NSGAII}(\mathcal{M}, s_\diamond)$ subject to $\mathbb{E}[\boldsymbol{g}(\mathbf{x}, s_\diamond) \mid \mathcal{D}] > 0$

---

$\mathbf{k}(\boldsymbol{x}, \boldsymbol{x}_{1:N}) = [k(\boldsymbol{x}, \boldsymbol{x}_1) \ \ldots \ k(\boldsymbol{x}, \boldsymbol{x}_N)]^\top$ representing an $(NM) \times 1$ column vector with each $k(\boldsymbol{x}, \boldsymbol{x}_i)$ representing an $M \times M$ covariance matrix, and $\boldsymbol{\Sigma} = [k(\boldsymbol{x}_i, \boldsymbol{x}_j)]_{i,j=1}^N$ is the $(NM) \times (NM)$ Gram matrix.

### 4.2. Causal Hypervolume Knowledge-Gradient

We extend the HVKG approach (Daulton et al., 2023) by integrating causal information into the acquisition to infer a hypervolume-maximizing Pareto set at $s_\diamond$. The optimization algorithm proceeds in rounds (hereafter referred to as iterations). At each iteration $t$, the acquisition function selects a configuration–fidelity pair $(\mathbf{x}, s)$ for intervention.

**Definition 4.1.** Given a finite Pareto frontier $\mathcal{P} \subset \mathbb{R}^M$ and a reference point $\boldsymbol{r} \in \mathbb{R}^M$, the hypervolume (HV) is the volume of the dominated space[5]: $\text{HV}(\mathcal{P}, \boldsymbol{r}) = \lambda(\bigcup_{\mathbf{y}' \in \mathcal{P}} \{\mathbf{y} \in \mathbb{R}^M : \mathbf{y}' \preceq \mathbf{y} \preceq \boldsymbol{r}\})$, where $\lambda$ is the Lebesgue measure.

Let $\mathcal{D}_{(\mathbf{x},s)} := \mathcal{D} \cup \{(\mathbf{x}, s, \mathbf{y})\}$ denote the dataset augmented with an additional observation $(\mathbf{x}, s, \mathbf{y})$, and $\bar{\mathcal{X}} \subseteq \mathcal{X}$ denote the feasible region satisfying constraints for all $q \in [Q]$. We define $\mu_\diamond(\mathbf{x}, \mathcal{D}) := \{\mathbb{E}_\mathcal{D}[\boldsymbol{f}(\mathbf{x}, s_\diamond)]\}_{\mathbf{x} \in \mathbf{x}}$ as the MF-CGP posterior mean at the target fidelity for candidate configurations[6] $\mathbf{x} \in \bar{\mathcal{X}}$, $\text{HV}_{\text{GP}}[\mathcal{D}_{(\mathbf{x},s)}] := \text{HV}[\mu_\diamond(\mathbf{x}|\mathcal{D}_{(\mathbf{x},s)})]$ as the hypervolume from the GP posterior mean, and $\text{HV}_{\text{CI}}[\mathcal{D}_{(\mathbf{x},s)}] := \text{HV}[\boldsymbol{f}_{\text{do}}(\mathbf{x}, s_\diamond)]$ as the hypervolume from causal estimates. The Causal Hypervolume Knowledge-

---

[5]For brevity, $\boldsymbol{r}$ is omitted from the reminder of the paper.

[6]The candidate configuration samples can be generated using techniques such as Latin Hypercube samples, and Sobol sequence.

Gradient (C-HVKG) acquisition function is defined as:

$$\mathcal{A}(\mathbf{x}, s) = \frac{1}{c(\mathbf{x}, s)} \mathbb{E}_\mathcal{D}\left[\max_{\mathbf{x} \subseteq \mathcal{X}} \text{HV}[\mathcal{D}_{(\mathbf{x},s)}] - \nu_\diamond^*\right], \text{ with}$$

$$\text{HV}\left[\mathcal{D}_{(\mathbf{x},s)}\right] = \text{HV}_{\text{GP}}\left[\mathcal{D}_{(\mathbf{x},s)}\right] + w \cdot \text{HV}_{\text{CI}}\left[\mathcal{D}_{(\mathbf{x},s)}\right],$$

where $\nu_\diamond^* := \max_{\mathbf{x} \in \mathcal{X}} \text{HV}[\mathcal{D}]$ and $w \in [0, 1]$ balances the contribution from the MF-CGP posterior (learned from $\mathcal{D}$) and the CPM estimates (learned from $\widehat{\mathcal{D}}$). Conceptually, C-HVKG quantifies the expected increase in the hypervolume of the Pareto frontier at the target fidelity by combining the predictive uncertainty of the MF-CGP with the causal knowledge encoded in the CPM, which is particularly beneficial when $\mathcal{D}$ is sparse. At each iteration $t$, we select the next $(\mathbf{x}_t, s_t)$ by maximizing the C-HVKG:

$$(\mathbf{x}_t, s_t) = \mathop{\arg\max}_{\substack{(\mathbf{x},s) \in \mathcal{X} \times \mathcal{S} \\ \mathbb{E}[\boldsymbol{g}(\mathbf{x},s_\diamond)|\mathcal{D}]>0}} \mathcal{A}(\mathbf{x}, s). \tag{2}$$

Although C-HVKG cannot be computed analytically, following Daulton et al. (2023, Theorem B.1), we obtain an unbiased estimator by approximating the outer expectation via Monte Carlo:

$$\widehat{\mathcal{A}}(\mathbf{x}, s) = \frac{1}{c(\mathbf{x}, s)}\left[\frac{1}{N}\sum_{i=1}^N\left(\max_{\mathbf{x} \subseteq \mathcal{X}} \text{HV}\left[\mathcal{D}_{(\mathbf{x},s)}^i\right]\right) - \nu_\diamond^*\right],$$

where $\mathcal{D}_{(\mathbf{x},s)}^i = \mathcal{D} \cup \{(\mathbf{x}, s, \mathbf{y}^i)\}$ with each $\mathbf{y}^i$ a realization or "fantasy" sample from the MF-CGP posterior predictive distribution $p(\mathbf{y}|\mathbf{x}, s, \mathcal{D})$. For each fantasy $\mathbf{y}^i$, the updated MF-CGP posterior mean can be computed analytically. Notably, causal estimates $\boldsymbol{f}_{\text{do}}(\mathbf{x}, s_\diamond)$ remain constant across fantasies, as they are learned from $\widehat{\mathcal{D}}$ and do not depend on $\mathcal{D}$. Thus, only $\text{HV}_{\text{GP}}$ is updated with each fantasy sample, while $\text{HV}_{\text{CI}}$ is computed once.

We present the complete RESCUE algorithm in Algorithm 1. To initialize $\mathcal{D}_0$, we use a cost-aware sampling strategy described in Appendix B.1 (Algorithm 2). The time complexity of Algorithm 1 has three main components: (i) initial CPM construction on $\mathbf{V}$ from $\widehat{\mathcal{D}}$ using PC or DirectLiNGAM, which requires $\mathcal{O}(|\widehat{\mathcal{D}}| |\mathbf{V}|^i)$ for PC, where $i$ is the maximum degree of $\mathbf{V}$ in $\mathcal{G}$, or $\mathcal{O}(|\widehat{\mathcal{D}}| |\mathbf{V}|^3 + |\mathbf{V}|^4)$ for DirectLiNGAM; (ii) causal inference via do-calculus for computing $\boldsymbol{f}_{\text{do}}(\mathbf{x}, s)$, with complexity $\mathcal{O}(|\mathbf{V}|^2 + |\widehat{\mathcal{D}}|)$ per query; and (iii) MF-CGP inference per iteration, with complexity $\mathcal{O}((NM)^3)$ to construct and invert the Gram matrix. The CPM is updated every $N_l$ iteration. The space complexity is $\mathcal{O}((NM)^2)$ for storing the Gram matrix, where $N$ is the number of observations.

## 5. Theoretical Analysis

We analyze RESCUE's performance when auxiliary sources poorly approximate the target fidelity, extending the single-objective regret bounds from Mikkola et al. (2023) to the

multi-objective setting. RESCUE's cumulative hypervolume regret is governed by two factors: (i) the standard GP confidence bound that shrinks with data collection, and (ii) the misspecification between the true objective and the causal prior computed via do-calculus from the learned CPM. We show that the regret bound remains finite regardless of the cross-fidelity bias magnitude, ensuring robustness even when auxiliary sources are arbitrarily unreliable. We provide the proofs in Appendix A.

**Definition 5.1.** The cumulative hypervolume regret after $T$ iterations is $\sum_{t=1}^{T}(\mathrm{HV}^* - \mathrm{HV}(\mathcal{P}_t))$, where $\mathrm{HV}^* = \mathrm{HV}(\mathcal{P}^*)$ is the true hypervolume at the target fidelity and $\mathcal{P}_t$ is the Pareto-front approximation at iteration $t$.

Following assumptions are required for our analysis. Assumptions 5.2–5.3 are standard in GP-based BO, while Assumptions 5.4–5.5 concern the multi-fidelity causal setting.

**Assumption 5.2** (GP regularity). For all $(m, q) \in [M] \times [Q]$, $f_m(\mathbf{x}, s)$ and $h_q(\mathbf{x}, s)$ lies in a Reproducing Kernel Hilbert Space (RKHS) with a bounded, Lipschitz continuous kernel.

**Assumption 5.3** (HV Lipschitz continuity). The hypervolume indicator is Lipschitz continuous: $\exists\,\mathcal{L} > 0$ such that for any two Pareto fronts $\mathcal{P}_1, \mathcal{P}_2$, $|\mathrm{HV}(\mathcal{P}_1) - \mathrm{HV}(\mathcal{P}_2)| \leq \mathcal{L} \cdot \max_{\mathbf{y} \in \mathcal{P}_1 \cup \mathcal{P}_2} \|\mathbf{y}_1 - \mathbf{y}_2\|$ where $\mathbf{y}_1 \in \mathcal{P}_1$ and $\mathbf{y}_2 \in \mathcal{P}_2$ are corresponding points.

**Assumption 5.4** (Agnostic causal prior approximation). There exists a function $\boldsymbol{f}_{\mathrm{do}} : \mathcal{X} \times \mathcal{S} \to \mathbb{R}^M$ such that: (i) for each $s \in \mathcal{S}$, every component of $\boldsymbol{f}_{\mathrm{do}}(\cdot, s)$ lies in the RKHS of Assumption 5.2 with norm at most $B > 0$; and (ii) for some $\xi \geq 0$, $\sup_{(\mathbf{x}, s) \in \mathcal{X} \times \mathcal{S}} \|\boldsymbol{f}(\mathbf{x}, s) - \boldsymbol{f}_{\mathrm{do}}(\mathbf{x}, s)\| \leq \xi$. In other words, the true objective $\boldsymbol{f}$ is uniformly $\xi$-close (in $L_\infty$ norm) to an RKHS function $\boldsymbol{f}_{\mathrm{do}}$ with bounded norm.

**Assumption 5.5** (Cost structure). Without loss of generality, we normalize costs so that $c(\mathbf{x}, s_\diamond) = 1$, and there exists $c_{\min} > 0$ such that $c(\mathbf{x}, s) \geq c_{\min}$ for all $(\mathbf{x}, s) \in \mathcal{X} \times \mathcal{S}$.

The following lemmas establish key theoretical results.

**Lemma 5.6** (Misspecified MF-CGP confidence bound). Under Assumptions 5.2 and 5.4, the causal prior error $\xi$ additively inflates the standard GP confidence bound. Specifically, for any $\rho \in (0, 1)$, there exists a nondecreasing sequence $\{\beta_t\}_{t \geq 1}$ such that, with probability at least $1 - \rho$, for all $\mathbf{x} \in \mathcal{X}$ and all $t \geq 1$,

$$\|\boldsymbol{f}(\mathbf{x}, s_\diamond) - \boldsymbol{\mu}_t(\mathbf{x}, s_\diamond)\| \leq \beta_t^{1/2} \|\boldsymbol{\sigma}_t(\mathbf{x}, s_\diamond)\| + \xi,$$

where $\beta_t$ is a confidence parameter depending on $B$, the information gain, and the noise level, as in the misspecified GP-UCB analyses of Bogunovic & Krause (2021) and Chowdhury & Gopalan (2017).

**Lemma 5.7** (HV approximation bound). Let $\bar{\mathcal{X}} \subseteq \mathcal{X}$ denote the set of truly feasible configurations. Let $\mathcal{P}^*$ be the true Pareto front of $\boldsymbol{f}$ over $\bar{\mathcal{X}}$, and write $\mathrm{HV}^* := \mathrm{HV}(\mathcal{P}^*)$ for its HV. Define the surrogate-induced Pareto front $\mathcal{P}_t = \mathrm{Pareto}(\{\boldsymbol{\mu}_t(\mathbf{x}, s_\diamond) : \mathbf{x} \in \bar{\mathcal{X}}\})$. Under Assumption 5.3, there exists a constant $\mathcal{L} > 0$ such that the HV approximation error is bounded by the maximum prediction error:

$$\mathrm{HV}^* - \mathrm{HV}(\mathcal{P}_t) \leq \mathcal{L} \cdot \max_{\mathbf{x} \in \bar{\mathcal{X}}} \|\boldsymbol{f}(\mathbf{x}, s_\diamond) - \boldsymbol{\mu}_t(\mathbf{x}, s_\diamond)\|.$$

**Lemma 5.8** (One-step acquisition comparison). Let $(\mathbf{x}_t, s_t)$ is selected by C-HVKG at iteration $t$, and let $\Delta_t^{SF}(\mathbf{x}_t)$ denote the expected hypervolume improvement that would be achieved by evaluating $\mathbf{x}_t$ at the target fidelity $s_\diamond$ under the true objective function $\boldsymbol{f}$:

$$\Delta_t^{SF}(\mathbf{x}_t) = \mathbb{E}\left[\mathrm{HV}(\mathcal{P}_{t+1}^{SF}) - \mathrm{HV}(\mathcal{P}_t^{SF}) \,\big|\, \mathbf{x}_t, s_\diamond, \mathcal{D}_{0:t-1}\right].$$

Then the cost-weighted acquisition value of RESCUE satisfies

$$c(\mathbf{x}_t, s_t)\,\mathcal{A}(\mathbf{x}_t, s_t) \geq \Delta_t^{SF}(\mathbf{x}_t) - \mathcal{L}\left(\max_{\mathbf{x} \in \bar{\mathcal{X}}} \beta_t^{1/2} \|\boldsymbol{\sigma}_t(\mathbf{x}, s_\diamond)\| + \xi\right). \tag{3}$$

**Theorem 5.9** (RESCUE cumulative HV regret). Under Assumptions 5.2, 5.4, and 5.3, fix any horizon $T \geq 1$. Then for any $\rho \in (0, 1)$, there exists a nondecreasing sequence $\{\beta_t\}_{t \geq 1}$ such that, with probability at least $1 - \rho$,

$$\sum_{t=1}^{T}\left(\mathrm{HV}^* - \mathrm{HV}(\mathcal{P}_t)\right) \leq \mathcal{L}\sum_{t=1}^{T}\left(\beta_t^{1/2} \max_{\mathbf{x} \in \bar{\mathcal{X}}} \|\boldsymbol{\sigma}_t(\mathbf{x}, s_\diamond)\| + \xi\right), \tag{4}$$

*Proof sketch.* Lemma 5.7 bounds $\mathrm{HV}^* - \mathrm{HV}(\mathcal{P}_t)$ by the maximum prediction error over the feasible region $\bar{\mathcal{X}}$. Lemma 5.6 provides a uniform bound on the prediction error for each configuration, with the error controlled by $\beta_t^{1/2} \|\boldsymbol{\sigma}_t(\mathbf{x}, s_\diamond)\| + \xi$, where the first term captures GP uncertainty and the second term quantifies the causal misspecification. Combining these two lemmas and maximizing over configurations yields the per-iteration bound. Summing over all iterations $t = 1, \ldots, T$ produces the cumulative regret bound in Eq. (4). See Appendix A.4 for the complete proof.

An immediate consequence of Theorem 5.9 addresses the critical concern of unreliable auxiliary sources.

**Corollary 5.10** (Unreliable auxiliary source robustness). Suppose Assumption 5.4 holds, and let $\boldsymbol{\delta}(\mathbf{x}, s) := \boldsymbol{f}(\mathbf{x}, s) - \boldsymbol{f}(\mathbf{x}, s_\diamond)$ denote the cross-fidelity bias of auxiliary sources at fidelity $s \neq s_\diamond$. Assume that the causal prior $\boldsymbol{f}_{\mathrm{do}}(\mathbf{x}, s_\diamond)$ is learned from observational data $\widehat{\mathcal{D}}$ via an appropriate causal identification strategy (e.g., do-calculus), so that the approximation error $\xi$ depends only on the quality and size of $\widehat{\mathcal{D}}$ and not on $\boldsymbol{\delta}$.

Then, for any horizon $T$ and any $\rho \in (0, 1)$, the RESCUE regret bound of Theorem 5.9 holds with the same constants independently of the magnitude of the cross-fidelity bias $\boldsymbol{\delta}$. In particular, for any sequence of auxiliary sources with $\sup_{\mathbf{x}, s \neq s_\diamond} \|\boldsymbol{\delta}(\mathbf{x}, s)\| \to \infty$, the regret bound remains finite and unchanged.

# 6. Experiments

We evaluate RESCUE in a variety of synthetic and real-world problems to assess its effectiveness in finding optimal configurations within a given budget. Our evaluation seeks to answer the following Research Questions (RQs): (1) To what extent RESCUE improves sample efficiency and optimization performance under a fixed budget compared to the baselines? (2) How does the relative performance of RESCUE change with problem dimensionality and the presence of constraints? (3) How does RESCUE allocate evaluations across fidelities compared to the baselines, and does this lead to cost-effective use of low- and high-fidelity information sources? See Appendix B for full experimental details[7].

## 6.1. Experimental settings

**Baselines** We compare RESCUE against qEHVI (Daulton et al., 2020) (non-MF), MOMF (Irshad et al., 2024), and MF-HVKG (Daulton et al., 2023). We use BoTorch implementations (Balandat et al., 2020) for all baselines. For qEHVI, we treat $s$ as a design variable.

**Evaluation metric** Following the evaluation criterion of Daulton et al. (2023), we obtain an approximate Pareto set by optimizing the MF-CGP posterior means with NSGA-II. We then evaluate the true objectives at these configurations and compute the HV dominated by the resulting true Pareto frontier, referred to as the inferred HV. We report the mean and standard deviation over 20 independent runs of each experiment to evaluate the trade-off between cost and log HV regret. The log HV regret is defined as the logarithm of the difference between the maximum HV of the true Pareto frontier and the inferred HV.

**Synthetic Problems** We consider two synthetic MF test functions from Irshad et al. (2024): (i) Branin-Currin ($d = 2, M = 2$) and (ii) Park ($d = 4, M = 2$).

**Real-World Problems** We consider five real-world problem classes spanning robotics, ML, and Healthcare. (i) **Robot Navigation** ($d = 26, M = 2, Q = 2$) from Hossen et al. (2025) involves tuning planner, controller, and costmap parameters for *Turtlebot 4* navigation in a complex environment. The target system operates with obstacles and narrow passages, while two sources consist of a *Turtlebot 4* in an obstacle-free setting and a *Gazebo* simulation (Fig. 1). The task is deemed successful if the robot reaches three goal locations and dock autonomously to a charging station, and the objectives are evaluated under constraints of collision risk $< 5.5$ and task completion rate $\geq 0.8$. For reproducibility, we curate a benchmark by collecting

---

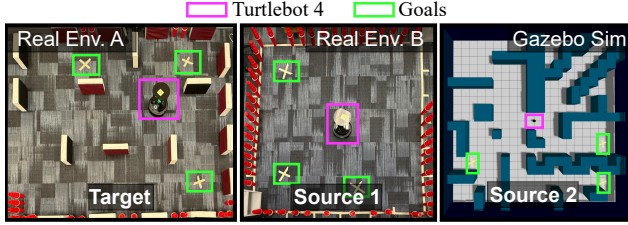

Figure 1. Real-robot environments with and without obstacles (left and middle, respectively), and a *Gazebo* simulation (right).

real-robot samples and provide a surrogate model derived from these data (Appendix B.5). (ii) **XGBoost HPO** ($d = 13, M = 2$) and (iii) **Ranger HPO** ($d = 5, M = 3$) use YAHPO Gym (Pfisterer et al., 2022), where fidelity is determined by the number of training samples. (iv) **Constrained XGBoost HPO** ($d = 13, M = 2, Q = 1$) and (v) **Constrained Ranger HPO** ($d = 5, M = 2, Q = 1$) are constrained variants from the same benchmark. (vi) **Healthcare** ($d = 3, M = 2, Q = 1$), following the setup in Aglietti et al. (2020), we consider optimizing statin use and Prostate Specific Antigen (PSA) while enforcing a clinical constraint that cancer risk $< 0.3$. For all experiments, the cost function is defined as $c(\mathbf{x}, s) = \exp(4.8s)$, so that for a fixed $s$ all configurations incur the same cost, leaving configuration-dependent cost modeling to future work. We assume the DAG is unknown and learn the CPM from observational data sampled randomly from each domain. When the DAG is known, this observational data collection can be skipped.

## 6.2. Results

Figure 2 shows that RESCUE significantly improves sample efficiency and optimization performance compared to the baselines. In synthetic problems (Fig. 2a, 2b), both HVKG and qEHVI perform strongly and closely track RESCUE, which suggests that in low dimensional and unconstrained settings, even SF MOBO can already be effective. As dimensionality increases and constraints are enforced (Fig. 2c–2h), RESCUE exhibits consistently larger gains. HVKG remains competitive on many problems but is dominated by RESCUE on all real-world and constrained ones and is even outperformed by qEHVI on the constrained Ranger HPO problem (Fig. 2h). MOMF often fails to identify any feasible configurations on the constrained problems, consistent with the acquisition function misalignment reported by Daulton et al. (2023) where fidelity is treated as an additional objective. Overall, RESCUE delivers consistent performance across the low and high dimensional, constrained and unconstrained problems. We attribute these gains to the causal prior encoded in the MF-CGP, which captures the casual relationships between and across configurations, fidelities, objectives and constraints, and thereby allows RESCUE to systematically exploit cross-fidelity generalization and to focus evaluations on causally promising regions of the

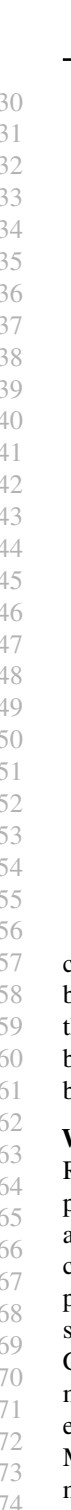
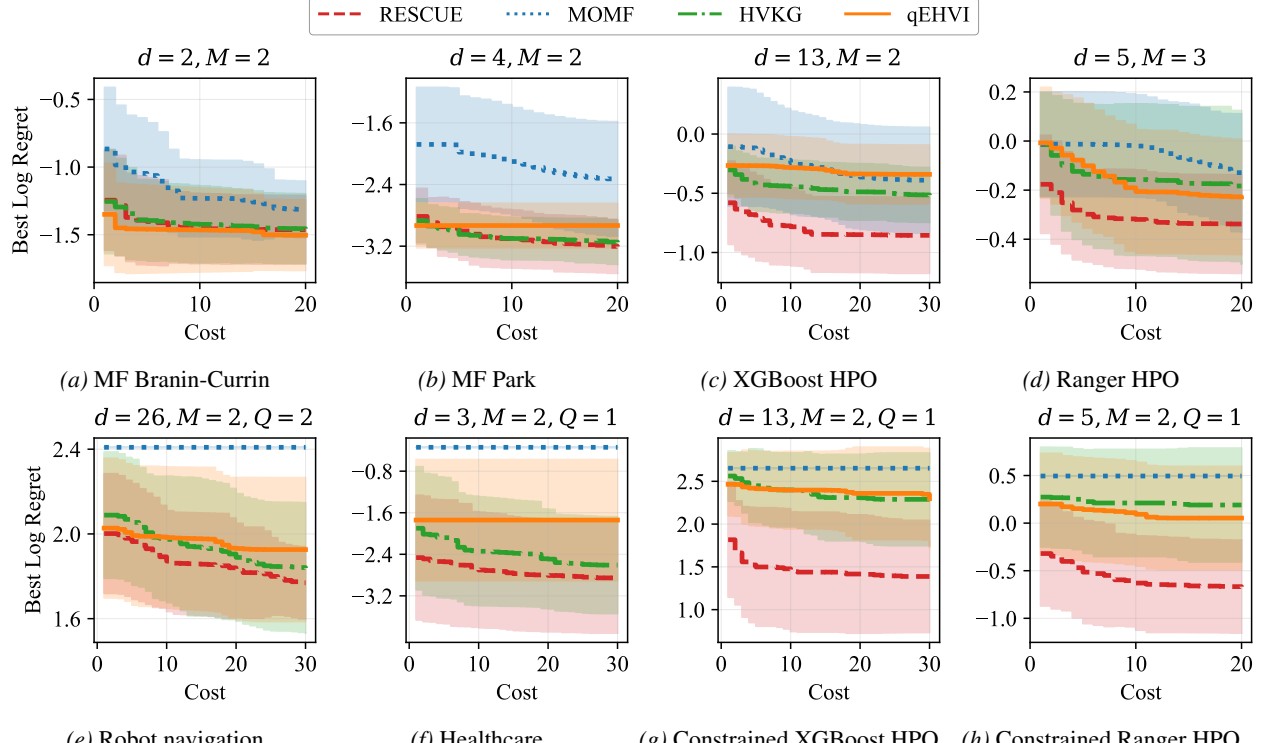

*Figure 2.* Cumulative intervention cost vs. best log HV regret, where "best" denotes the minimum regret achieved up to that cost.

configuration space, leading to effective use of the available budget. These results address RQ1 and RQ2 by showing that RESCUE improves sample efficiency under a fixed budget and that its performance remains stable as problems become higher dimensional and constrained.

**Why does RESCUE work better?** To understand why RESCUE achieves superior optimization performance compared to the baselines, we examine how it uses the budget and handles constraints. Figure 3 shows that RESCUE uses cheaper fidelities more often than the baselines. HVKG performs on par with RESCUE, as both methods implement similar acquisition functions, but it never surpasses RESCUE. In contrast, qEHVI and MOMF consume the budget much faster with high fidelity queries. This behavior is expected for qEHVI (a non-MF method) and explains why MOMF performs similarly to qEHVI despite being an MF method. Figure 4 shows the constraint violation rate (violations per iteration). RESCUE achieves the lowest violation rate across constrained problems. MOMF rarely finds feasible configurations, consistent with the acquisition-function misalignment discussed earlier. Surprisingly, qEHVI yields lower violations than HVKG except in Fig. 4b. By exploiting causal structure, RESCUE systematically selects configuration-fidelity pairs that balance cost and information and avoid infeasible regions, which explains its lower regret in Fig. 2 and addresses RQ3.

## 7. Discussion

To the best of our knowledge, this is the first work to consider causal calculus in a multi-fidelity BO setting. The proposed design is general and extendable to other problems. To apply RESCUE to a novel problem, the practitioner must identify (i) configuration options, (ii) performance metrics, and (iii) key performance indicators. The abstraction level of the variables in the CPM is chosen by the practitioner and can range from high-level system options down to the hardware level.

The NP-hard complexity of causal discovery (Chickering et al., 2004) implies that the identified causal model may not always represent the ground-truth causal relationships among variables. It is therefore important to recognize potential discrepancies between the discovered causal structure and the actual one. However, such causal models can still be employed to achieve better performance compared to ML-based approaches in systems optimization (Dubslaff et al., 2022) and debugging tasks (Hossen et al., 2023), because causal models aim to avoid capturing spurious correlations (Glymour et al., 2019).

The CPM shapes both the GP prior mean and the covariance, introducing a strong inductive bias. When this prior is misspecified or overly confident, it can misguide the GP posterior toward incorrect interventional effects and cause

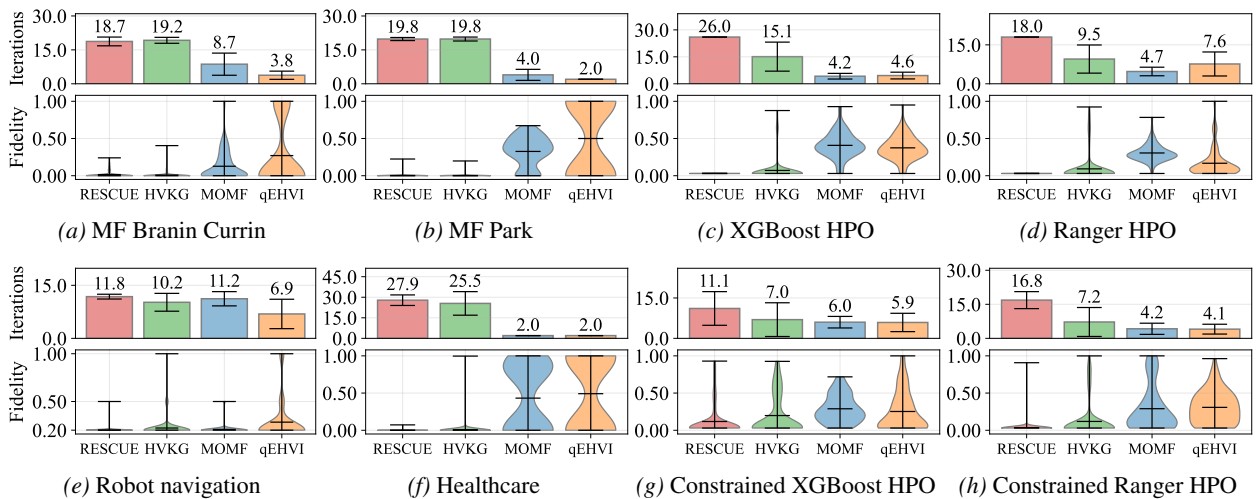

*Figure 3.* Fidelity queries and iteration counts under a fixed budget. For each problem, the top panel reports the number of iterations achieved by each method, and the bottom panel shows the empirical distribution of queried fidelities. Each violin shows the fidelity distribution with a vertical min–max range line and a mean marker.

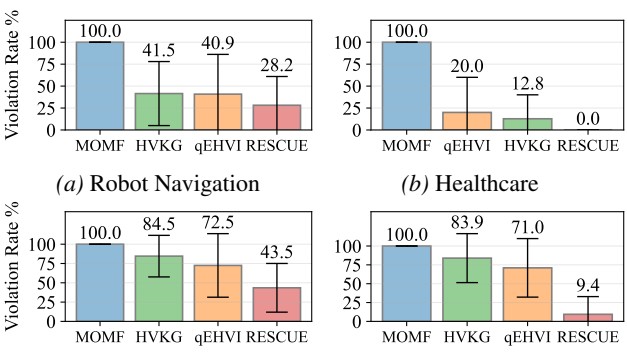

*Figure 4.* Constraint violation rate percentage.

an ill-conditioned covariance that requires additional regularization (e.g., added diagonal jitter). In practice, the causal prior should *guide* rather than *dominate*, and the GP must retain sufficient flexibility in its hyperparameters to correct or override erroneous causal information.

Our implementation relies on DoWhy (Blöbaum et al., 2024) for causal inference. To enable PyTorch (Paszke et al., 2019) integration and GPU acceleration, we train a neural-network proxy in PyTorch on interventional samples generated from the CPM. This approximation can introduce additional bias if the proxy does not faithfully represent the underlying causal mechanisms. Directly calling DoWhy at each optimization step would avoid this proxy, but was empirically too slow in our setting due to the lack of GPU support.

An immediate extension of RESCUE is to use causal effect estimates to adaptively reduce the configuration space, by identifying options with negligible causal effect on the performance objectives and focusing the search on causally

relevant configuration options, which may further improve sample efficiency in high dimensional problems (Hossen et al., 2025).

## 8. Conclusion

We present RESCUE, a multi-fidelity, Multi-objective Bayesian Optimization (MOBO) method that integrates causal calculus into the multi-fidelity setting. RESCUE learns a causal performance model (CPM) that captures causal relations between configuration options, fidelities, objectives, and constraints. The CPM parameterizes the prior mean and covariance of a multi-output multi-fidelity GP over objectives and constraints. Building on this model, we designed and implemented a multi-fidelity causal hypervolume knowledge-gradient acquisition strategy that balances cost-information trade-off. Our theoretical analysis shows that RESCUE's cumulative hypervolume regret can be bounded to its full-fidelity (target only) MOBO analog, and that this bound holds even when auxiliary sources are unreliable. Empirically, across synthetic and real-world problems in robotics, machine learning, and healthcare, RESCUE systematically selects configuration-fidelity pair that balance cost and information gain, leading to lower hypervolume regret and fewer constraint violations than baseline methods. These results indicate that integrating causal calculus into multi-fidelity MOBO is a promising direction for more sample-efficient optimization with improved constraint handling and causal interpretability in complex systems.

## Impact Statement

This paper presents work whose goal is to advance the field of Machine Learning. There are many potential societal consequences of our work, none which we feel must be specifically highlighted here.

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

## Notations

*Table 1.* Mathematical notations

| Description | Notation |
| --- | --- |
| Configuration space | $\mathcal{X}$ |
| Feasible configuration space | $\bar{\mathcal{X}} \subseteq \mathcal{X}$ |
| Fidelity space | $\mathcal{S}$ |
| Configuration | $\mathbf{x} \in \mathcal{X}$ |
| Configuration-fidelity pair | $\boldsymbol{x} := (\mathbf{x}, s)$ |
| Fidelity | $s \in \mathcal{S}$ |
| Target fidelity | $s_\diamond \in \mathcal{S}$ |
| Number of configuration options | $d$ |
| Number of objectives | $M$ |
| Number of constraints | $Q$ |
| Objective value | $\mathbf{y}$ |
| Constraint thresholds | $\gamma_q, q \in Q$ |
| Total budget | $\Lambda$ |
| Total number of iterations | $T$ |
| Pareto frontier at target fidelity | $\mathcal{P}$ |
| Directed acyclic graph (DAG) | $\mathcal{G}$ |
| Causal performance model (CPM) | $\mathcal{C}$ |
| Observational dataset | $\widehat{\mathcal{D}}$ |
| Interventional dataset | $\mathcal{D}$ |
| CPM error | $\xi$ |
| Lipschitz constant | $\mathcal{L}$ |

## A. Proofs

### A.1. Proof of Lemma 5.6

The inclusion of Lemma 5.6 is critical for establishing the theoretical validity of using *estimated* causal priors within a Gaussian Process bandit framework. Standard regret analyses typically rely on a strong "realizable" assumption that the true objective function lies exactly within the Reproducing Kernel Hilbert Space (RKHS) associated with the GP prior. In RESCUE, however, the causal prior is derived from observational data or simplified physical models, and is therefore an *approximation* rather than a perfect oracle. By adopting the *agnostic* GP setting, Lemma 5.6 rigorously quantifies how the causal misspecification error $\xi$ propagates into the posterior predictions.

Moreover, this result provides the key guarantee underpinning the exploration–exploitation behavior of the algorithm under model mismatch. As long as the causal prior is a reasonable approximation (i.e., $\xi$ is bounded), the true objective function remains contained within the inflated confidence intervals with high probability. This directly connects our causal inference component to the robust GP bandit literature (e.g., Bogunovic & Krause (2021)), and shows that RESCUE does not require a perfectly specified causal graph to succeed, but instead degrades gracefully as a function of the causal estimation error.

*Proof.* We work in the standard "agnostic" or misspecified GP setting. By Assumption 5.4, there exists an RKHS function $\boldsymbol{f}_{\mathrm{do}}$ with $\|\boldsymbol{f}_{\mathrm{do}}(\cdot, s)\|_{\mathcal{H}_k} \leq B$ for all $s$ and such that

$$\boldsymbol{f}(\mathbf{x}, s) = \boldsymbol{f}_{\mathrm{do}}(\mathbf{x}, s) + \delta(\mathbf{x}, s), \qquad \sup_{(\mathbf{x}, s)} \|\delta(\mathbf{x}, s)\| \leq \xi.$$

The observations collected by the optimizer are

$$y_t \ = \ \boldsymbol{f}(\mathbf{x}_t, s_t) + \xi_t \ = \ \boldsymbol{f}_{\mathrm{do}}(\mathbf{x}_t, s_t) + \big(\delta(\mathbf{x}_t, s_t) + \xi_t\big),$$

where $\{\xi_t\}$ is the observation noise process from Assumption 5.2. Thus, from the perspective of GP regression on the latent function $\boldsymbol{f}_{\mathrm{do}}$, the effective noise is $\tilde{\xi}_t := \delta(\mathbf{x}_t, s_t) + \xi_t$. Since $\delta$ is uniformly bounded and $\xi_t$ is sub-Gaussian, $\tilde{\xi}_t$ remains sub-Gaussian (with shifted mean) and satisfies the tail assumptions required for the concentration results cited below. The MF-CGP posterior mean $\boldsymbol{\mu}_t$ and variance $\boldsymbol{\sigma}_t$ are then the usual kernel ridge regression estimators based on these observations and the kernel in Assumption 5.2.

Under the RKHS norm bound on $\boldsymbol{f}_{\mathrm{do}}$ and the uniform control $\|\delta(\mathbf{x}, s)\| \leq \xi$, results from the misspecified GP bandit literature (see, e.g., Lemma 3 in Chowdhury & Gopalan (2017) and the EC-GP-UCB analysis of Bogunovic & Krause (2021); cf. also Srinivas et al. (2009) for the realizable case) show that the misspecification error propagates as an additive inflation of the usual GP confidence width. Concretely, there exists a sequence $\{\beta_t\}_{t \geq 1}$ such that, for any $\rho \in (0, 1)$, with probability at least $1 - \rho$, for all $\mathbf{x} \in \bar{\mathcal{X}}$ and all $t \geq 1$,

$$\big\|\boldsymbol{f}(\mathbf{x}, s_\diamond) - \boldsymbol{\mu}_t(\mathbf{x}, s_\diamond)\big\| \ \leq \ \beta_t^{1/2} \big\|\boldsymbol{\sigma}_t(\mathbf{x}, s_\diamond)\big\| \ + \ \xi. \tag{5}$$

Intuitively, the term $\beta_t^{1/2} \|\boldsymbol{\sigma}_t\|$ is the standard GP confidence width that would arise if $\boldsymbol{f}$ itself lay in the RKHS (the realizable case), while the uniform misspecification error $\xi$ accounts for the irreducible approximation error $\delta$. Eq. (5) is precisely the claimed inequality, which completes the proof. $\qquad\square$

*Remark* A.1. The key difference from the realizable analysis of Srinivas et al. (2009) is that the MF-CGP posterior $\boldsymbol{\mu}_t$ now concentrates around the true objective $\boldsymbol{f}$ only up to an additive bias controlled by $\sup_{(\mathbf{x}, s)} \|\boldsymbol{f}(\mathbf{x}, s) - \boldsymbol{f}_{\mathrm{do}}(\mathbf{x}, s)\|$. Lemma 5.6 makes this bias explicit via the term $\xi$, yielding an "enlarged" confidence tube in the spirit of the EC-GP-UCB algorithm of Bogunovic & Krause (2021).

### A.2. Proof of Lemma 5.7

*Proof.* Let

$$A := \{\boldsymbol{f}(\mathbf{x}, s_\diamond) : \mathbf{x} \in \bar{\mathcal{X}}\}, \qquad B_t := \{\boldsymbol{\mu}_t(\mathbf{x}, s_\diamond) : \mathbf{x} \in \bar{\mathcal{X}}\}.$$

By definition of the hypervolume indicator, dominated points do not contribute, so

$$\mathrm{HV}^* = \mathrm{HV}(\mathcal{P}^*) = \mathrm{HV}(A), \qquad \mathrm{HV}(\mathcal{P}_t) = \mathrm{HV}(B_t).$$

Assumption 5.3 states that the hypervolume indicator is Lipschitz-continuous with respect to perturbations of the objective values. In particular, there exists $\mathcal{L} > 0$ such that

$$\big|\mathrm{HV}(A) - \mathrm{HV}(B_t)\big| \ \leq \ \mathcal{L} \cdot \sup_{\mathbf{x} \in \bar{\mathcal{X}}} \big\|\boldsymbol{f}(\mathbf{x}, s_\diamond) - \boldsymbol{\mu}_t(\mathbf{x}, s_\diamond)\big\|.$$

Since $\mathrm{HV}^* = \mathrm{HV}(A)$ and $\mathrm{HV}(\mathcal{P}_t) = \mathrm{HV}(B_t)$, this yields

$$\big|\mathrm{HV}^* - \mathrm{HV}(\mathcal{P}_t)\big| \ \leq \ \mathcal{L} \cdot \max_{\mathbf{x} \in \bar{\mathcal{X}}} \big\|\boldsymbol{f}(\mathbf{x}, s_\diamond) - \boldsymbol{\mu}_t(\mathbf{x}, s_\diamond)\big\|.$$

Finally, for any real number $x$ we have $x \leq |x|$, hence

$$\mathrm{HV}^* - \mathrm{HV}(\mathcal{P}_t) \ \leq \ \big|\mathrm{HV}^* - \mathrm{HV}(\mathcal{P}_t)\big| \ \leq \ \mathcal{L} \cdot \max_{\mathbf{x} \in \bar{\mathcal{X}}} \big\|\boldsymbol{f}(\mathbf{x}, s_\diamond) - \boldsymbol{\mu}_t(\mathbf{x}, s_\diamond)\big\|,$$

which completes the proof. $\qquad\square$

### A.3. Proof of Lemma 5.8

*Proof.* By definition of the C-HVKG selection rule, $(\mathbf{x}_t, s_t)$ maximizes the cost-normalized acquisition function. Equivalently, it maximizes the cost-weighted value $c(\mathbf{x}, s) \mathcal{A}(\mathbf{x}, s)$, so for any $\mathbf{x}$,

$$c(\mathbf{x}_t, s_t) \mathcal{A}(\mathbf{x}_t, s_t) \ \geq \ c(\mathbf{x}, s_\diamond) \mathcal{A}(\mathbf{x}, s_\diamond). \tag{6}$$

In particular, taking $\mathbf{x} = \mathbf{x}_t$ yields

$$c(\mathbf{x}_t, s_t)\, \mathcal{A}(\mathbf{x}_t, s_t) \;\geq\; c(\mathbf{x}_t, s_\diamond)\, \mathcal{A}(\mathbf{x}_t, s_\diamond).$$

Next, we relate $\mathcal{A}(\mathbf{x}_t, s_\diamond)$ to the true expected improvement $\Delta_t^{\mathrm{SF}}(\mathbf{x}_t)$. The quantity $\mathcal{A}(\mathbf{x}_t, s_\diamond)$ is the expected hypervolume improvement *per unit cost* computed under the MF-CGP predictive distribution; hence $c(\mathbf{x}_t, s_\diamond)\, \mathcal{A}(\mathbf{x}_t, s_\diamond)$ is the model-based expected hypervolume improvement. In contrast, $\Delta_t^{\mathrm{SF}}(\mathbf{x}_t)$ is the corresponding improvement under the true objective $\boldsymbol{f}$ at $s_\diamond$.

Applying the hypervolume approximation bound (Lemma 5.7) to the pre- and post-update Pareto fronts induced by $\boldsymbol{\mu}_t$ and by $\boldsymbol{f}$, we obtain

$$\left| c(\mathbf{x}_t, s_\diamond)\, \mathcal{A}(\mathbf{x}_t, s_\diamond) - \Delta_t^{\mathrm{SF}}(\mathbf{x}_t) \right| \;\leq\; \mathcal{L} \cdot \max_{\mathbf{x} \in \bar{\mathcal{X}}} \left\| \boldsymbol{f}(\mathbf{x}, s_\diamond) - \boldsymbol{\mu}_t(\mathbf{x}, s_\diamond) \right\|.$$

Using the general inequality $A \geq B - |A - B|$ with $A = c(\mathbf{x}_t, s_\diamond)\, \mathcal{A}(\mathbf{x}_t, s_\diamond)$ and $B = \Delta_t^{\mathrm{SF}}(\mathbf{x}_t)$, we deduce

$$c(\mathbf{x}_t, s_\diamond)\, \mathcal{A}(\mathbf{x}_t, s_\diamond) \;\geq\; \Delta_t^{\mathrm{SF}}(\mathbf{x}_t) \;-\; \mathcal{L} \cdot \max_{\mathbf{x} \in \bar{\mathcal{X}}} \left\| \boldsymbol{f}(\mathbf{x}, s_\diamond) - \boldsymbol{\mu}_t(\mathbf{x}, s_\diamond) \right\|.$$

Finally, substituting the misspecified MF-CGP confidence bound from Lemma 5.6, which guarantees

$$\left\| \boldsymbol{f}(\mathbf{x}, s_\diamond) - \boldsymbol{\mu}_t(\mathbf{x}, s_\diamond) \right\| \;\leq\; \beta_t^{1/2} \left\| \boldsymbol{\sigma}_t(\mathbf{x}, s_\diamond) \right\| + \xi \qquad \forall\, \mathbf{x} \in \bar{\mathcal{X}},$$

and combining this with Eq. (6) (with $\mathbf{x} = \mathbf{x}_t$) yields the inequality in (3), completing the proof. $\qquad\square$

## A.4. Proof of Theorem 5.9

*Proof.* For each iteration $t$, Lemma 5.7 (hypervolume approximation bound) applied to the RESCUE surrogate implies

$$\mathrm{HV}^* - \mathrm{HV}(\mathcal{P}_t) \;\leq\; \mathcal{L} \cdot \max_{\mathbf{x} \in \bar{\mathcal{X}}} \left\| \boldsymbol{f}(\mathbf{x}, s_\diamond) - \boldsymbol{\mu}_t(\mathbf{x}, s_\diamond) \right\|.$$

Under Assumptions 5.2 and 5.4, the misspecified MF-CGP confidence bound of Lemma 5.6 guarantees that, for any $\rho \in (0, 1)$, there exists a sequence $\{\beta_t\}_{t \geq 1}$ such that, with probability at least $1 - \rho$,

$$\left\| \boldsymbol{f}(\mathbf{x}, s_\diamond) - \boldsymbol{\mu}_t(\mathbf{x}, s_\diamond) \right\| \;\leq\; \beta_t^{1/2} \left\| \boldsymbol{\sigma}_t(\mathbf{x}, s_\diamond) \right\| + \xi \qquad \forall\, \mathbf{x} \in \bar{\mathcal{X}}, \; \forall\, t \geq 1.$$

Combining the two inequalities and maximizing over $\mathbf{x} \in \bar{\mathcal{X}}$ yields, on the same high-probability event,

$$\mathrm{HV}^* - \mathrm{HV}(\mathcal{P}_t) \;\leq\; \mathcal{L} \left( \beta_t^{1/2} \max_{\mathbf{x} \in \bar{\mathcal{X}}} \left\| \boldsymbol{\sigma}_t(\mathbf{x}, s_\diamond) \right\| + \xi \right), \qquad t = 1, \dots, T.$$

Summing over $t = 1, \dots, T$ gives

$$\sum_{t=1}^{T} \left( \mathrm{HV}^* - \mathrm{HV}(\mathcal{P}_t) \right) \;\leq\; \mathcal{L} \sum_{t=1}^{T} \left( \beta_t^{1/2} \max_{\mathbf{x} \in \bar{\mathcal{X}}} \left\| \boldsymbol{\sigma}_t(\mathbf{x}, s_\diamond) \right\| + \xi \right),$$

which is exactly Eq. (4). This establishes the claimed high-probability regret bound for RESCUE. $\qquad\square$

## A.5. Proof of Corollary 5.10

*Proof.* By Theorem 5.9, under Assumptions 5.2, 5.4, and 5.3, we have, with probability at least $1 - \rho$,

$$\sum_{t=1}^{T} \left( \mathrm{HV}^* - \mathrm{HV}(\mathcal{P}_t) \right) \;\leq\; \mathcal{L} \sum_{t=1}^{T} \left( \beta_t^{1/2} \max_{\mathbf{x} \in \bar{\mathcal{X}}} \left\| \boldsymbol{\sigma}_t(\mathbf{x}, s_\diamond) \right\| + \xi \right).$$

By construction of the MF-CGP and Assumption 5.4, the term $\xi$ is defined as a uniform bound

$$\sup_{(\mathbf{x}, s) \in \mathcal{X} \times \mathcal{S}} \left\| \boldsymbol{f}(\mathbf{x}, s) - \boldsymbol{f}_{\mathrm{do}}(\mathbf{x}, s) \right\| \;\leq\; \xi,$$

where $\boldsymbol{f}_{\mathrm{do}}$ is the interventional causal prior learned from observational data $\widehat{\mathcal{D}}$. The key point is that $\boldsymbol{f}_{\mathrm{do}}(\mathbf{x}, s_\diamond)$ is identified from the *target-fidelity intervention* $(s = s_\diamond)$ and does not require modeling, or relying on the auxiliary fidelities $s \neq s_\diamond$. Thus, under the stated causal assumptions, the approximation error $\xi$ depends only on CPM learning error (e.g., finite-sample error in $\widehat{\mathcal{D}}$ and functional approximation error) and is independent of the cross-fidelity bias $\boldsymbol{\delta}(\mathbf{x}, s)$. The remaining term in the regret bound,

$$\sum_{t=1}^{T} \beta_t^{1/2} \max_{x \in \mathcal{X}} \|\sigma_t(x, s^\diamond)\|,$$

depends on the GP posterior variance at the target fidelity, which in standard GP regression is independent of the observed function values and determined only by the kernel, query locations, and noise level. Hence, although the magnitude of the true cross-fidelity bias $\delta(x, s)$ may affect the observed auxiliary outputs, it does not affect $\sigma_t(x, s^\diamond)$ and therefore does not enter the bound.

Consequently, for any family of auxiliary sources where $\sup_{\mathbf{x}, s \neq s_\diamond} \|\boldsymbol{\delta}(\mathbf{x}, s)\| \to \infty$, the right-hand side of the inequality in Theorem 5.9 remains unchanged. In particular, the same constants $\mathcal{L}$, $\{\beta_t\}$, and $\xi$ apply, and the bound does not blow up as $\|\boldsymbol{\delta}\| \to \infty$. This establishes the claimed robustness to unreliable auxiliary sources. $\square$

### A.6. Discussion of Theorem 5.9

Theorem 5.9 establishes that RESCUE's cumulative hypervolume regret is controlled by two interpretable terms: (i) the GP posterior uncertainty at the target fidelity, and (ii) the causal misspecification error $\xi$.

**Role of causal accuracy.** When the CPM learned from observational data $\widehat{\mathcal{D}}$ is accurate, the causal prior computed via do-calculus has small error ($\xi$ small), and RESCUE's regret is dominated by the GP uncertainty term, which decreases with data accumulation. In this case, RESCUE can perform more iterations via cheaper lower-fidelity evaluations under the same budget constraint. This additional exploration opportunity can be beneficial when sources are informative, though the benefit depends on the actual fidelity choices made by the C-HVKG acquisition function. Conversely, when observational data is limited or the causal structure is complex (leading to large $\xi$), the bound ensures controlled degradation, preventing unbounded regret accumulation.

**Independence from cross-fidelity bias.** A key insight from Corollary 5.10 is that the regret bound does not depend on the cross-fidelity bias $\boldsymbol{\delta}(\mathbf{x}, s)$. Since the causal prior $\boldsymbol{f}_{\mathrm{do}}(\mathbf{x}, s_\diamond)$ is computed using do-calculus rather than by extrapolating across fidelities, $\xi$ depends only on the quality of $\widehat{\mathcal{D}}$, not on how poorly auxiliary sources approximate the target. This ensures robustness, even when simulators or cheaper proxies are misaligned with the target system. The worst-case regret bound remains controlled by $\xi$ and GP uncertainty, not by simulation-to-reality gaps.

## B. Experiment Details

### B.1. Cost-Aware Initial Sampling

To initialize $\mathcal{D}_0 = \{(\mathbf{x}_i, s_i, \mathbf{y}_i)\}_{i=1}^{N}$ (line 1 of Algorithm 1), we sample configurations and fidelities using Algorithm 2, where the inverse CDF is:

$$\mathrm{CDF}(s) = \begin{cases} \dfrac{\int_0^s c(\mathbf{x}, \tilde{s})^{-1} d\tilde{s}}{\int_0^1 c(\mathbf{x}, \tilde{s})^{-1} d\tilde{s}} & \text{(continuous fidelities)} \\[2ex] \displaystyle\sum_{j=1}^{J} \dfrac{c(\mathbf{x}, s_j)^{-1}}{\sum_{i=1}^{S} c(\mathbf{x}, s_i)^{-1}} & \text{(discrete fidelities)}, \end{cases} \tag{7}$$

and $p(s) \propto c(\mathbf{x}, s)^{-1}$, so that lower-cost fidelities are sampled more frequently, increasing sample diversity under the initialization budget. We assume that for a fixed $s$, all $\mathbf{x} \in \mathcal{X}$ incur the same cost, leaving configuration-dependent cost modeling to future work.

### B.2. Synthetic problems

**Branin-Currin** We use the modified multi-fidelity, multi-objective Branin-Currin function from Irshad et al. (2024). The configuration space is given in Table 4a, and the discovered DAG is shown in Fig. 7a.

---

**Algorithm 2** Cost-Aware Initial Sampling

---

**Require:** Sampling budget $\Lambda_{\text{init}}$, cost function $c(\cdot, \cdot)$, objective function $f(\cdot, \cdot)$
**Ensure:** Initial dataset $\mathcal{D}_0 = \{(\mathbf{x}_i, s_i, \mathbf{y}_i)\}_{i=1}^N$
1: Initialize $\mathcal{D}_0 = \emptyset$, $C_0 = 0$, $i = 0$
2: **while** $C_i < \Lambda_{\text{init}}$ **do**
3:     $i \leftarrow i + 1$
4:     Sample configuration $\mathbf{x}_i \sim \mathcal{U}(\mathcal{X})$
5:     Sample fidelity $s_i = \text{CDF}^{-1}(u)$ using Eq. (7), where $u \sim \mathcal{U}(0, 1)$
6:     **if** $C_{t-1} + c(\mathbf{x}_i, s_t) \leq \Lambda_{\text{init}}$ **then**
7:         Evaluate objectives $\mathbf{y}_i = f(\mathbf{x}_t, s_t)$
8:         $\mathcal{D}_{0:i} \leftarrow \mathcal{D}_{0:i-1} \cup \{(\mathbf{x}_i, s_i, \mathbf{y}_i)\}$
9:         $C_i \leftarrow C_{i-1} + c(\mathbf{x}_i, s_i)$
10:    **end if**
11: **end while**
12: **return** $\mathcal{D}_0$

---

**Park**  We use the modified multi-fidelity, multi-objective Park function from Irshad et al. (2024). The configuration space is given in Table 4b, and the discovered DAG is shown in Fig. 7b.

### B.3. Real-world problems

**XGBoost HPO**  We use the YAHPO Gym (Pfisterer et al., 2022) multi-fidelity, multi-objective HPO benchmark (Scenario: iaml_xgboost, Instance: 41146). The objectives are mean misclassification error (mmce) and memory footprint of the model (rammodel), both to be minimized. The configuration space is given in Table 4c and the discovered DAG is shown in Fig. 8.

**Ranger HPO**  We use the YAHPO Gym (Pfisterer et al., 2022) multi-fidelity, multi-objective HPO benchmark (Scenario: iaml_ranger, Instance: 1489). The objectives are mean misclassification error (mmce), number of features (nf), and interaction strength of features (ias), all to be minimized. The configuration space is given in Table 4c and the discovered DAG is shown in Fig. 8.

**Healthcare**  We cannot directly use the objective functions from Aglietti et al. (2020) because there is no multi-fidelity variant of this problem. Instead, we modify them to incorporate an additional fidelity input dimension:

$$\text{Statin} = \sigma(S \times (-13.0 + 0.1 \times \text{Age} + 0.2 \times \text{BMI})$$
$$\text{Cancer} = \sigma(S \times (2.2 - 0.05 \times \text{Age} + 0.01 \times \text{BMI} - 0.04 \times \text{Statin} + 0.2 \times \text{Aspirin}))$$
$$\text{PSA} = (S + 6.8)(0.04 \times \text{Age} - 0.15 \times \text{BMI} + 0.6 \times \text{Statin} + 0.55 \times \text{Aspirin} + \text{Cancer})$$

where $\sigma(\cdot)$ is the sigmoidal transformation defined as $\sigma(x) = \frac{1}{1+\exp(-x)}$, and Age $= 65$. The main difference is the addition of the fidelity parameter $S$, which recovers the original functions when $S = 1$. We aim to minimize Statin and PSA while ensuring Cancer $< 0.35$. The configuration space is given in Table 4e and the discovered DAG is shown in Fig. 12.

**Robot navigation**  Following the experimental setup of Hossen et al. (2025), we deploy mobile robots in a controlled indoor environment. The robots are operated using ROS 2 (Macenski et al., 2022) and the Nav2 (Macenski et al., 2020) navigation stack for localization, path planning, and trajectory tracking. We consider point-to-point navigation tasks and record navigation performance across fidelity levels. Specifically, we measure (i) Energy per meter $= \frac{\text{Energy}_{\text{total}}}{\text{Distance tranveled}}$, (ii) Task completion time, (iii) Task completion rate $= \frac{\sum \text{Tasks}_{\text{complted}}}{\sum \text{Tasks}}$, and (iv) a physics-based Collision-risk score defined as follows:

$$\text{Collision-risk score} = 10(0.2 \times r_{\text{speed}} + 0.3 \times r_{\text{safety}} + 0.2 \times r_{\text{raction}} + 0.1 \times r_{\text{perception}} + 0.15 \times r_{\text{goal}} + 0.05 \times r_{\text{sampling}}),$$

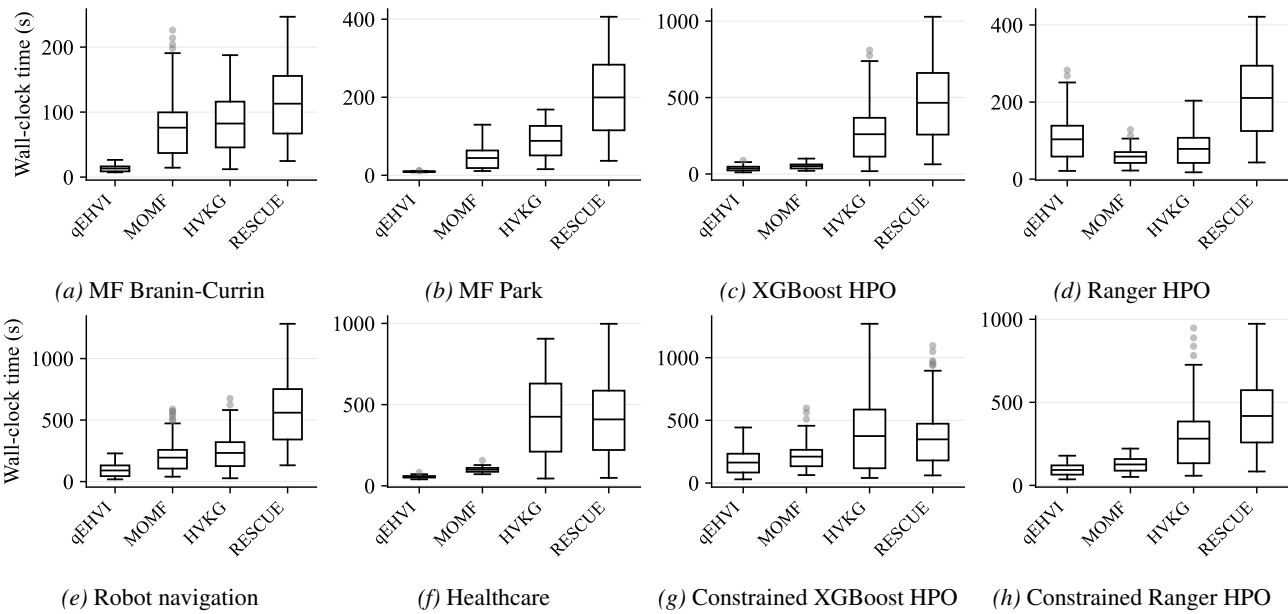

*Figure 5.* Optimization runtime (wall-clock time in seconds) for each problem and method. For every problem–method pair, we repeated the experiment with 20 independent seeds. Each boxplot summarizes the distribution of those 20 runs. The runtime includes the time required to obtain measurements across fidelities.

where

$$r_{\text{speed}} = \text{clip}\left(\frac{\text{speed\_decel\_ratio}}{2.0}, 0, 1\right), \text{ with speed\_decel\_ratio} = \frac{\text{max\_vel\_x}}{|\text{decel\_lim\_x}| + 0.01},$$

$$r_{\text{safety}} = \frac{1}{1 + \text{safety\_margin}}, \text{ with safety\_margin} = \frac{\text{local\_inflation\_radius}}{\text{max\_vel\_x} + 0.01},$$

$$r_{\text{reaction}} = \frac{1}{1 + \text{reaction\_margin}}, \text{ with reaction\_margin} = \frac{\text{local\_inflation\_radius}}{(\text{max\_vel\_x} \times \text{sim\_time}) + 0.01},$$

$$r_{\text{perception}} = \text{clip}\left(\frac{\text{perception\_risk}}{0.5}, 0, 1\right), \text{ with perception\_risk} = \frac{\text{local\_resolution}}{\text{local\_inflation\_radius} + 0.01},$$

$$r_{\text{goal}} = \text{clip}\left(\frac{\text{goal\_vs\_obstacle\_ratio}}{3.0}, 0, 1\right), \text{ with goal\_vs\_obstacle\_ratio} = \frac{\text{GoalAlign. scale} + \text{GoalDist. scale}}{\text{BaseObstacle. scale} + 0.01},$$

$$r_{\text{sampling}} = \frac{1}{1 + \frac{\text{sampling\_density}}{200}}, \text{ with sampling\_density} = \text{vx\_samples} \times \text{vtheta\_samples}.$$

Note that $\text{clip}(x, 0, 1) := \max(0, \min(x, 1))$. The configuration space is given in Table 4f and the discovered DAG is shown in Fig. 13.

**Constrained XGBoost HPO** We use the same setup as the XGBoost HPO problem with the goal of maximizing F1 score and minimizing training time (timetrain), while enforcing that the memory usage during training (ramtrain) is less than 15 MB. The configuration space is given in Table 4c and the discovered DAG is shown in Fig. 10.

**Constrained Ranger HPO** We use the same setup as the Ranger HPO problem with the goal of maximizing AUC (auc) and minimizing logloss, while enforcing timetrain < 1.3. The configuration space is given in Table 4d and the discovered DAG is shown in Fig. 11.

### B.4. Additional Results

**Runtime** All experiments were run on a single machine with an AMD Ryzen Threadripper 3960X CPU, an NVIDIA RTX A6000 GPU, 128 GB memory, and Ubuntu 20.04 with Python 3.10, Botorch 0.15.1 and DoWhy 0.13. Figure 5 reports

*Table 2.* Area-under-regret (AUR) curve comparison between RESCUE and baselines. Blue cells in the $p$-value column denote $p < 0.05$, and green/red cells in the Gain column indicate improvements / degradations of RESCUE over the corresponding baseline.

| Problem | Baseline | AUR $\downarrow$ | $p$-value | $t$-stat. | Cohen's $d$ | RESCUE Gain % | RESCUE AUR $\downarrow$ |
|---|---|---|---|---|---|---|---|
| Branin Currin $(d = 3, M = 2)$ | qEHVI | $-27.92 \pm 5.70$ | 5.50E$-$1 | 0.61 | 0.14 | $-2.74$ | |
| | MOMF | $-22.59 \pm 5.30$ | 9.07E$-$4 | $-4.00$ | $-0.94$ | 20.23 | $-27.16 \pm 5.33$ |
| | HVKG | $-26.83 \pm 5.50$ | 6.20E$-$1 | $-0.50$ | $-0.11$ | 1.24 | |
| Park $(d = 5, M = 2)$ | qEHVI | $-55.71 \pm 5.60$ | 7.93E$-$2 | $-1.85$ | $-0.41$ | 5.40 | |
| | MOMF | $-40.24 \pm 13.12$ | 5.96E$-$5 | $-5.13$ | $-1.147$ | 45.91 | $-58.72 \pm 6.89$ |
| | HVKG | $-58.45 \pm 5.21$ | 8.33E$-$1 | $-0.21$ | $-0.04$ | 0.46 | |
| XGBoost HPO $(d = 13, M = 2)$ | qEHVI | $-8.93 \pm 7.16$ | 1.62E$-$5 | $-5.82$ | $-1.34$ | 160.99 | |
| | MOMF | $-2.24 \pm 12.52$ | 5.23E$-$4 | $-4.21$ | $-0.97$ | 182.80 | $-23.31 \pm 9.21$ |
| | HVKG | $-13.39 \pm 6.57$ | 1.30E$-$3 | $-3.80$ | $-0.87$ | 74.14 | |
| Ranger HPO $(d = 5, M = 3)$ | qEHVI | $-3.16 \pm 4.45$ | 1.00E$-$1 | $-1.70$ | $-0.39$ | 86.07 | |
| | MOMF | $-0.89 \pm 3.89$ | 3.90E$-$4 | $-4.34$ | $-1.00$ | 561.02 | $-5.88 \pm 3.76$ |
| | HVKG | $-2.80 \pm 5.67$ | 6.00E$-$2 | $-2.00$ | $-0.46$ | 109.60 | |
| Robot navigation $(d = 26, M = 2, Q = 2)$ | qEHVI | $56.94 \pm 9.60$ | 2.60E$-$1 | $-1.16$ | $-0.26$ | 5.03 | |
| | MOMF | $69.86 \pm 1.46$E$-14$ | 6.76E$-$10 | $-11.33$ | $-2.53$ | 22.59 | $54.07 \pm 6.23$ |
| | HVKG | $56.20 \pm 8.20$ | 2.60E$-$1 | $-1.16$ | $-0.26$ | 3.79 | |
| Healthcare $(d = 3, M = 2, Q = 1)$ | qEHVI | $-50.49 \pm 34.60$ | 4.10E$-$2 | $-3.26$ | $-0.73$ | 56.76 | |
| | MOMF | $-9.88 \pm 1.82$E$-15$ | 1.10E$-$8 | $-9.55$ | $-2.14$ | 701.06 | $-79.14 \pm 32.43$ |
| | HVKG | $-69.12 \pm 28.38$ | 1.26E$-$1 | $-1.60$ | $-0.35$ | 14.50 | |
| Constrained XGBoost HPO $(d = 13, M = 2, Q = 1)$ | qEHVI | $69.20 \pm 14.25$ | 2.10E$-$5 | $-5.61$ | $-1.25$ | 39.13 | |
| | MOMF | $76.94 \pm 0.00$ | 2.27E$-$7 | $-7.84$ | $-1.75$ | 45.26 | $42.12 \pm 19.87$ |
| | HVKG | $68.26 \pm 13.23$ | 1.44E$-$5 | $-5.78$ | $-1.29$ | 38.29 | |
| Constrained Ranger HPO $(d = 5, M = 2, Q = 1)$ | qEHVI | $1.87 \pm 10.35$ | 1.60E$-$4 | $-4.75$ | $-1.09$ | 690.43 | |
| | MOMF | $9.40 \pm 1.83$E$-15$ | 2.11E$-$8 | $-9.45$ | $-2.17$ | 217.45 | $-11.04 \pm 9.43$ |
| | HVKG | $4.08 \pm 10.92$ | 1.03E$-$4 | $-4.95$ | $-1.14$ | 370.77 | |

the end-to-end optimization runtime, including the time to obtain measurements. Across the eight benchmarks, qEHVI and MOMF have the smallest mean runtimes, while RESCUE and HVKG are moderately slower but remain in the same order of magnitude. The higher runtime of RESCUE and HVKG arises from computing the (C-)HVKG acquisition with Monte Carlo fantasies for each batch. In RESCUE this cost is further increased by learning a CPM, and performing causal inference in the MF-CGP to construct the causal prior and in the C-HVKG acquisition.

**Experiment statistics**  Table 2 reports the area-under-regret (AUR) curve. For each problem and baseline, the AUR is obtained by integrating the instantaneous regret over the total budget using the trapezoidal rule. For each baseline, we report the paired $t$-test $p$-value, $t$-statistic, Cohen's $d$, and the corresponding relative AUR gain of RESCUE. Across the eight benchmarks, RESCUE almost always reduces AUR, with particularly large and statistically significant gains. Overall, averaging the Gain % column across problems, RESCUE achieve improvement by **130%** over qEHVI, **225%** over MOMF, and **77%** over HVKG.

### B.5. Multi-fidelity mobile robot navigation benchmark

We construct a three-fidelity benchmark inspired from the setup of Hossen et al. (2025). Data are collected from: (i) *Turtle-Bot4 in Gazebo simulation* ($s = 0.2$), (ii) *TurtleBot4 physical robot in an obstacle-free indoor environment* ($s = 0.5$), and (iii) *TurtleBot4 physical robot in a complex indoor environment* (target, $s = 1.0$). We collect 100 interventional samples at each fidelity by uniformly sampling configurations within the parameter bounds listed in Table 4f, yielding 300 samples in total. For all trials, the robot is deployed under ROS 2, and evaluated on five task metrics: task completion rate, task

*Table 3.* Robot navigation task surrogate model performance.

| Model output | Train | | Test | |
|---|---|---|---|---|
| | $R^2$ | RMSE | $R^2$ | RMSE |
| Task execution time (s) | 0.95 | 31.95 | 0.84 | 58.79 |
| Total energy (Wh) | 0.94 | 0.09 | 0.83 | 0.17 |
| Distance Traveled (m) | 0.97 | 3.28 | 0.73 | 8.39 |
| Collision risk score | 0.99 | 0.02 | 0.98 | 0.09 |
| Average | 0.97 | 8.83 | 0.81 | 16.86 |

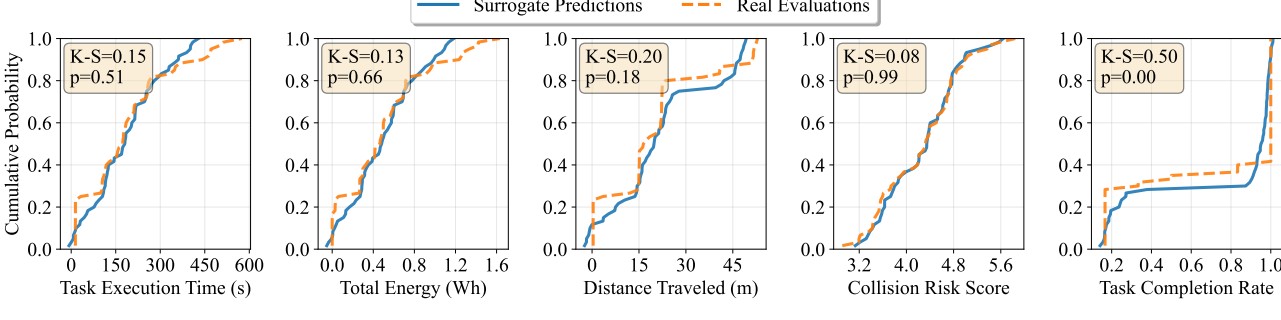

*(a)* Combined Empirical Cumulative Distribution Function (ECDF) for surrogate predictions and real evaluations.

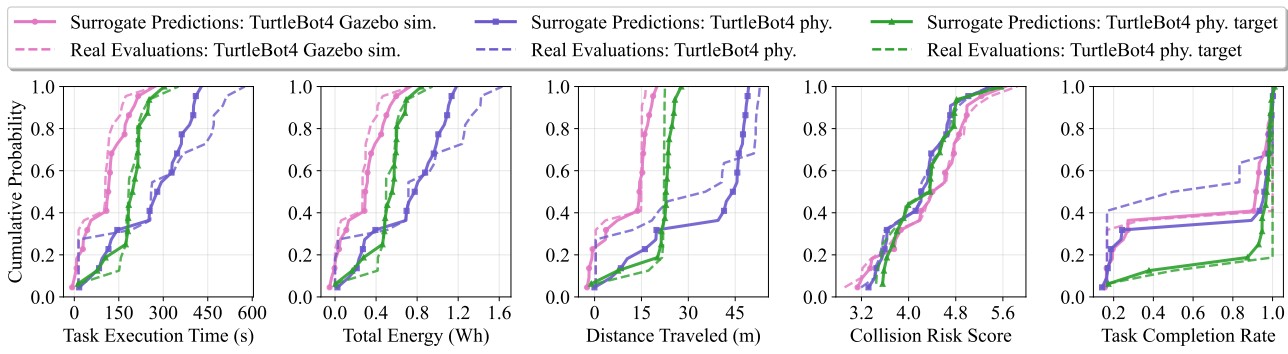

*(b)* Per-fidelity ECDF for surrogate predictions and real evaluations.

*Figure 6.* Surrogate model validation for the robot navigation task. Close alignment between predicted and real distributions, combined with consistent performance across simulation and physical robot fidelities, demonstrates the surrogate's effectiveness.

execution time, total energy consumption, distance traveled, and collision-risk score. We randomly select $80\%$ of the data for training and reserve $20\%$ for testing.

**Surrogate Model Training and Architecture**   We fit independent CatBoost (Prokhorenkova et al., 2018) gradient-boosted regressors for each metric using multi-output regression. Fidelity is included as a numeric covariate to retain its ordinal structure. During inference, we additionally derive energy-per-meter as the ratio of total energy to distance traveled. All trained models are exported to ONNX format for cross-platform compatibility.

**Distributional Alignment Across Fidelities**   Figure 5a reports the Empirical Cumulative Distribution Function (ECDF) of surrogate predictions and real evaluations aggregated across fidelities. The Kolmogorov-Smirnov statistic (K-S and $p$-values) indicates close alignment for most metrics. The largest deviation is for task completion rate. Because this metric is bounded in $[0, 1]$, its empirical distribution often concentrates near 0 or 1 across all fidelities. Such boundary concentration reduces variance and makes the K-S statistic more sensitive to small shifts between the surrogate and the real system, especially under the sample size of 100 per fidelity. Figure 5b disaggregates ECDFs by fidelity and shows that the surrogate reproduces fidelity-dependent shifts, confirming it captures cross-fidelity trends and fidelity-specific variability.

**Quantitative Performance** Table 3 reports train and test $R^2$ and Root Mean Square Error (RMSE) for each metric. On the test set, the surrogate achieves an average $R^2$ of 0.81. Energy and collision-risk have the smallest RMSE, distance traveled has a larger RMSE, and execution time has the largest RMSE. Combined with the distributional agreement in Fig. 6, the results indicate that the surrogate is sufficiently accurate to support the multi-fidelity mobile robot navigation benchmark, and any residual surrogate error propagates uniformly across all optimization methods including RESCUE and the baselines.

## Discovered DAGs and Configuration Space

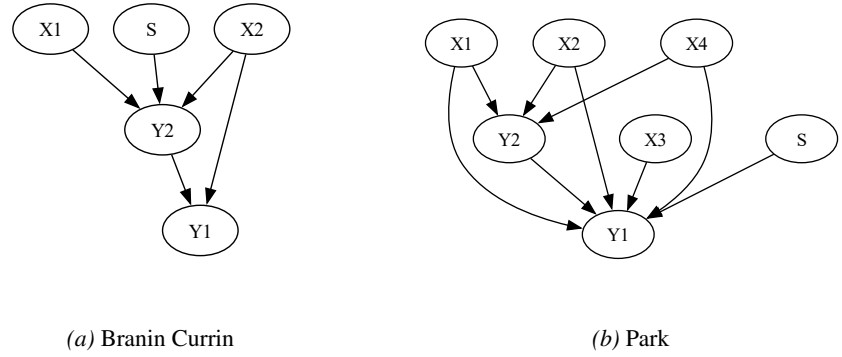

*(a)* Branin Currin                    *(b)* Park

*Figure 7.* Discovered DAG using PC algorithm.

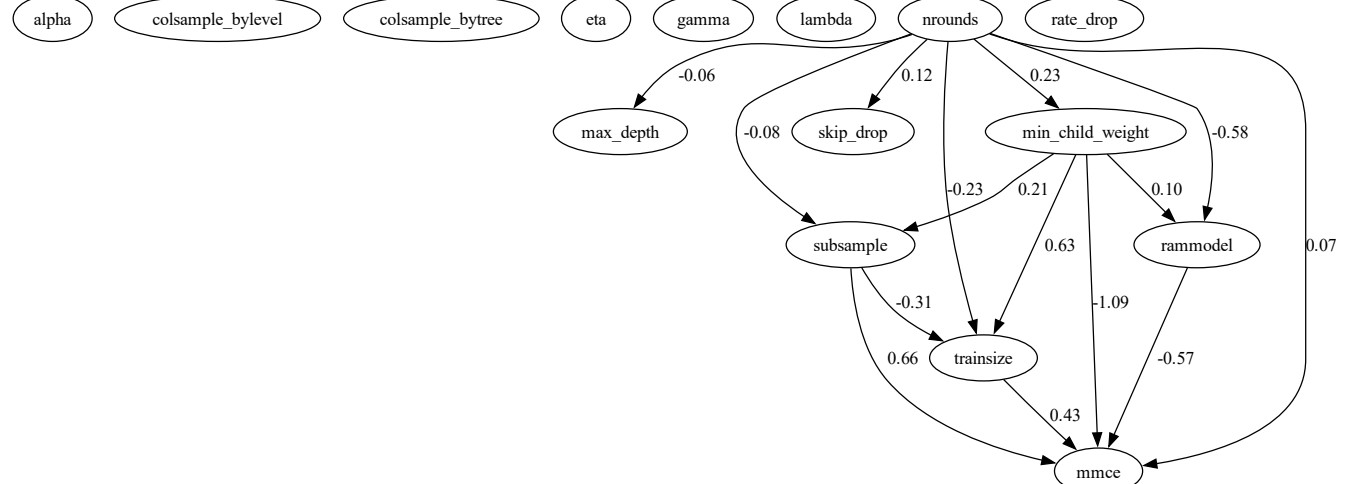

*Figure 8.* Discovered DAG of XGBoost HPO problem using DirectLiNGAM.

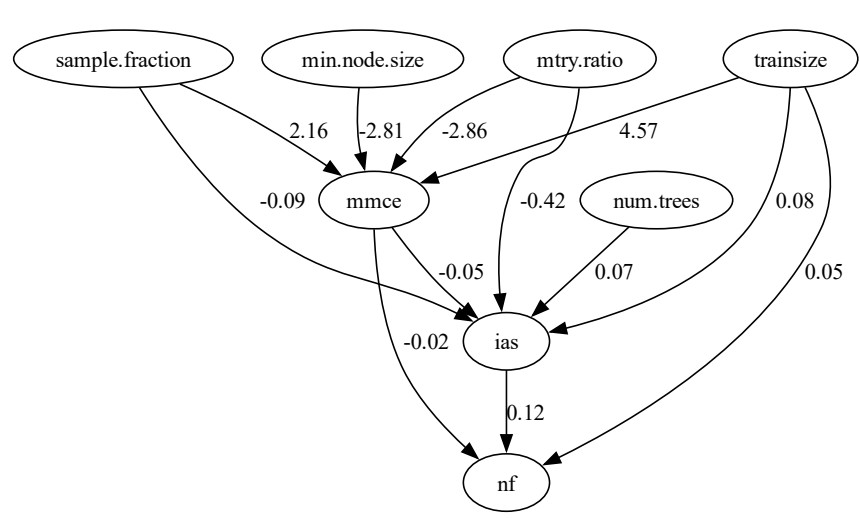

*Figure 9.* Discovered DAG of Ranger HPO problem using DirectLiNGAM.

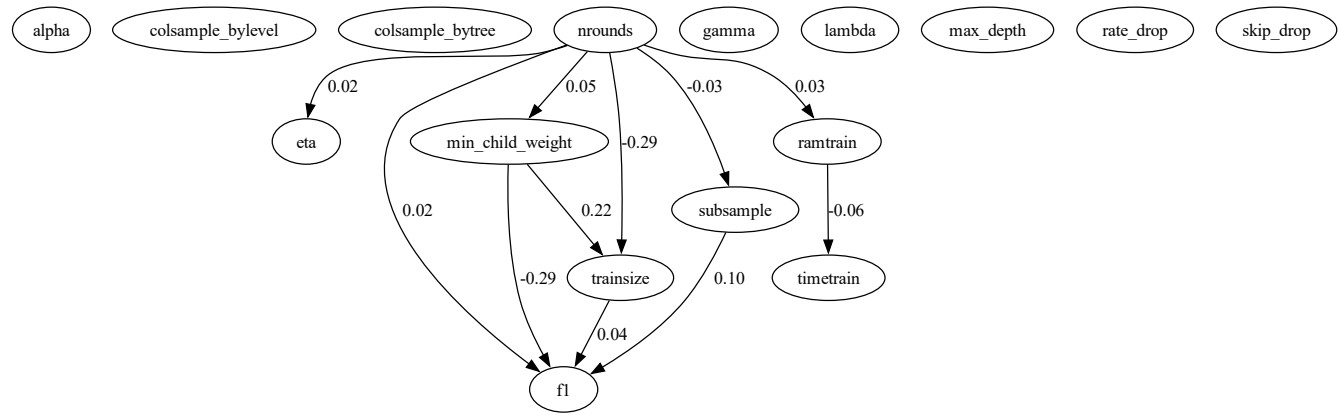

*Figure 10.* Discovered DAG of Constrained XGBoost problem HPO using DirectLiNGAM.

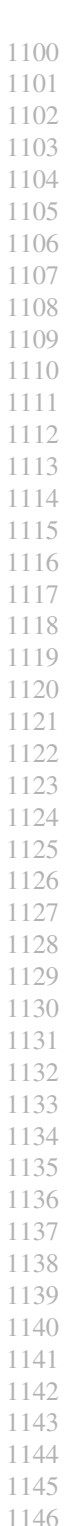

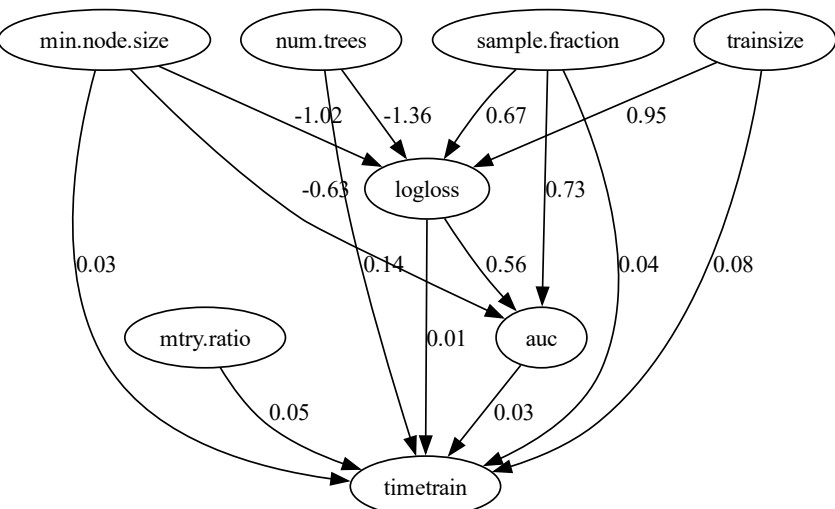

*Figure 11.* Discovered DAG of Constrained XGBoost HPO problem using DirectLiNGAM.

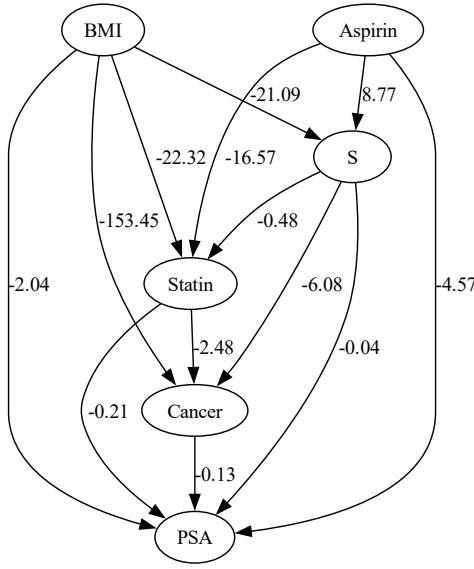

*Figure 12.* Discovered DAG of Healthcare problem using DirectLiNGAM.

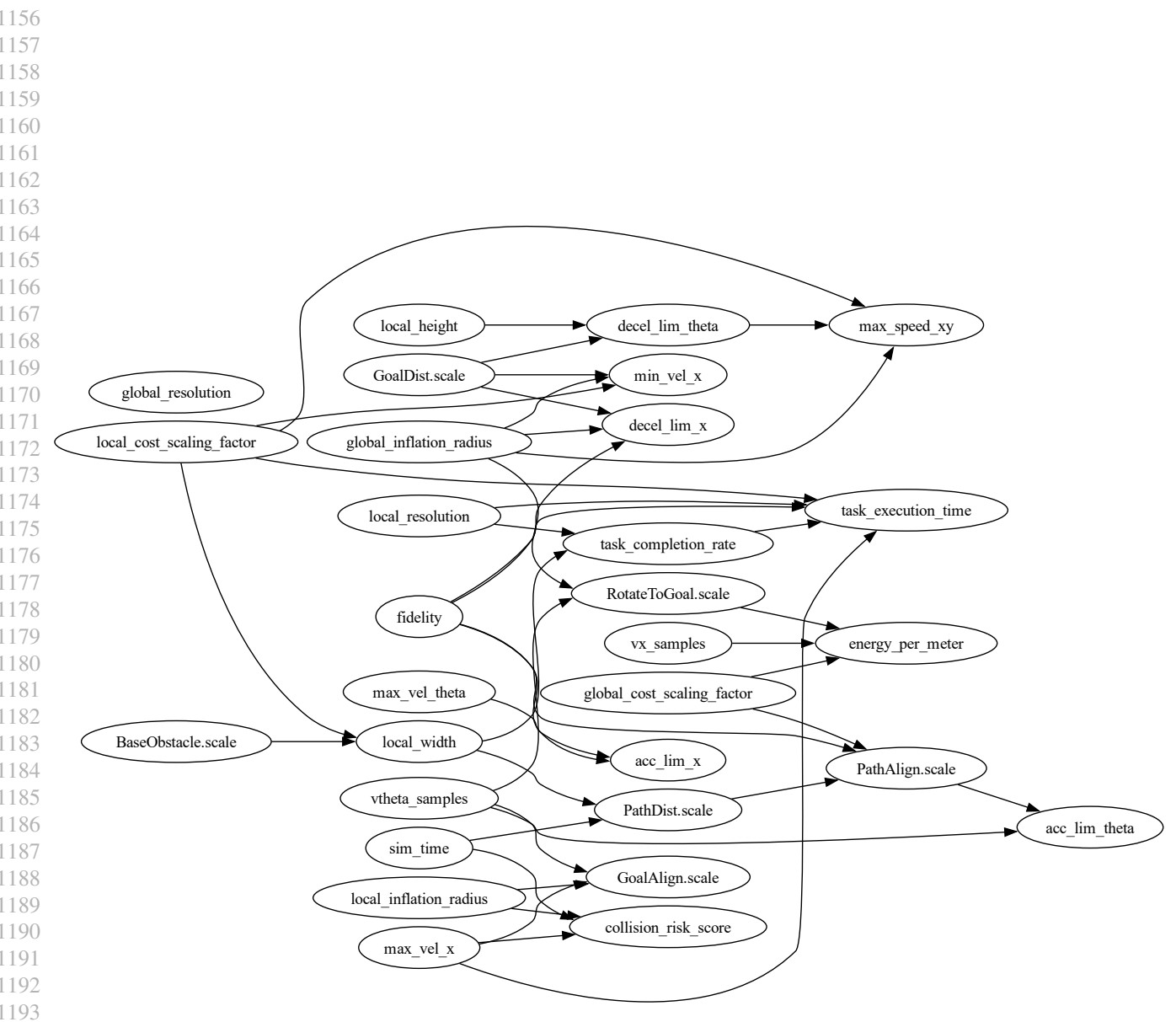

*Figure 13.* Discovered DAG of Robot navigation task problem using PC algorithm.

*Table 4.* Configuration Spaces of All Problems

*(a)* Branin–Currin

| Configuration options | Range |
|---|---|
| X1 | [0, 1] |
| X2 | [0, 1] |
| S | [0, 1] |

*(b)* Park

| Configuration options | Range |
|---|---|
| X1 | [0, 1] |
| X2 | [0, 1] |
| X3 | [0, 1] |
| X4 | [0, 1] |
| S | [0, 1] |

*(c)* XGBoost HPO (Scenario: iaml_xgboost, Instance: 41146)

| Configuration options | Range/Value |
|---|---|
| alpha | [0.0005, 1.0] |
| colsample_bylevel | [0.02, 1] |
| colsample_bytree | [0.01, 1] |
| eta | [0.0005, 1] |
| gamma | [0.0005, 1] |
| lambda | [0.0005, 1] |
| max_depth | [1, 15] |
| min_child_weight | [2.72, 149] |
| nrounds | [3, 2000] |
| rate_drop | [0, 1] |
| skip_drop | [0, 1] |
| subsample | [0.1, 1] |
| trainsize | [0.03, 1] |
| booster | dart |

*(d)* Ranger HPO (Scenario: iaml_ranger Instance: 1489)

| Configuration options | Range/Value |
|---|---|
| min.node.size | [1, 100] |
| mtry.ratio | [0, 1] |
| num.trees | [1, 2000] |
| sample.fraction | [0.1, 1] |
| trainsize | [0.03, 1] |
| replace | True |
| respect.unordered.factors | ignore |
| splitrule | gini |

*(e)* Healthcare

| Configuration options | Range |
|---|---|
| BMI | [20, 30] |
| Aspirin | [0, 1] |
| S | [0, 1] |

*(f)* Robot Navigation Task

| Configuration options | Range/Value |
|---|---|
| min_vel_x | [−3, 0] |
| max_vel_x | [0.1, 0.5] |
| max_vel_theta | [0.5, 2.0] |
| max_speed_xy | [0, 0.1] |
| acc_lim_x | [1.5, 4] |
| acc_lim_theta | [1.5, 5] |
| decel_lim_x | [−4, −1] |
| decel_lim_theta | [−5, −1.5] |
| vx_samples | [10, 40] |
| vtheta_samples | [10, 40] |
| sim_time | [1, 3] |
| BaseObstacle.scale | [0.01, 0.1] |
| PathAlign.scale | [10, 60] |
| GoalAlign.scale | [10, 60] |
| PathDist.scale | [10, 60] |
| GoalDist.scale | [10, 60] |
| RotateToGoal.scale | [10, 60] |
| local_width | [2, 5] |
| local_height | [2, 5] |
| local_resolution | [0.04, 0.1] |
| local_inflation_radius | [0.3, 0.6] |
| local_cost_scaling_factor | [2, 10] |
| global_resolution | [0.04, 0.1] |
| global_inflation_radius | [0.3, 0.6] |
| global_cost_scaling_factor | [2, 10] |
| xy_goal_tolerance | 0.25 |
| yaw_goal_tolerance | 0.25 |
| fidelity | 0.2, 0.5, 1.0 |

