# OpenReview forum: "Multi-Objective Multi-Fidelity Bayesian Optimization with Causal Priors"
_ICML.cc/2026/Conference — Submitted to ICML 2026_

### Official Review · Reviewer_8EHn · 2026-02-16

**Soundness:** 1
**Presentation:** 2
**Significance:** 2
**Originality:** 2
**Overall Recommendation:** 2
**Confidence:** 3

**Summary:**

The paper proposes a Causal Graph–Informed Multi-Objective Multi-Fidelity Bayesian Optimization framework with inequality constraints. The main contribution (and assumption) of the paper is that using a Structural Causal Model (SCM), derived from historical data \\( \\hat{D} \\) or specified by the user, can help solve Multi-Objective Multi-Fidelity Constrained Black-Box Optimization problems.

The paper uses a Multi-Fidelity Causal GP (MF-CGP) as a surrogate model, which is a multi-output GP with fidelity information embedded in the kernel computation and causal information incorporated from the obtained or specified SCM.

The paper proposes the Causal Hypervolume Knowledge Gradient (C-HVKG) acquisition function, which uses a weighted sum between hypervolume computation based on the MF-CGP and hypervolume computation from causal estimates.

**Compliance With Llm Reviewing Policy:**

Affirmed.

**Final Justification:**

I appreciate the authors' detailed rebuttal and the clarifications provided. However, my main concern remains: the full RESCUE method does not outperform the GP + C-HVKG ablation baseline, which raises questions about the effectiveness of the proposed __surrogate__. The empirical results do not convincingly demonstrate the benefit of the proposed component. I will therefore maintain my original score and thank the authors for their efforts.

**Key Questions For Authors:**

1. **Robustness to Misspecified SCM**

Could you comment on the case where the SCM is incorrectly defined (e.g., errors in causal discovery from the historical data)? The correctness of the SCM appears to be a strong assumption, and the behavior of the algorithm may heavily depend on the quality of the causal discovery step. If the inferred causal DAG is wrong, how robust is the proposed method? Is there any theoretical or empirical analysis of performance degradation under SCM misspecification?

2. **Use of NSGA-II in Algorithm 1**

The use of NSGA-II in line 14 of Algorithm 1 is somewhat confusing. It seems that C-HVKG is used for exploration and improving the surrogate model, while NSGA-II is used to obtain the final Pareto front. Why using points sampled from C-HVKG isnt enough?

3. **Inequality Constraints in Baselines**

How are inequality constraints implemented for the baseline methods? This detail should be clearly described in Section 6.1 to ensure fair comparison. Are constraints handled consistently across all methods ?

4. **Evaluation Metrics and Pareto Front Visualization**

While HV regret is a useful metric, it would also be helpful to include visualizations of the resulting Pareto fronts for 2D and 3D output cases. This would allow readers to qualitatively assess the quality of the obtained Pareto front compared to the ground truth. Additionally, other commonly used metrics such as Generational Distance (GD) and spread (diversity) metrics could provide a more comprehensive evaluation. Is there a reason these metrics were not included?

5. **Ground-Truth Pareto Set Approximation (Line 298)**

Please correct me if I am mistaken, but in line 298 it is stated that the ground-truth Pareto set is obtained by optimizing the MF-CGP posterior. However, in Daulton et al. (2023), the recommended approach is to compute an approximate Pareto set using NSGA-II on \\( \\mathbb{E}\[f(x)\] \\), where \\( f \\) is the true objective function. In your case, this would typically correspond to the objective evaluated at the target fidelity, rather than the MF-CGP surrogate.

If the MF-CGP posterior is indeed used to construct the ground-truth Pareto front across experiments, this could potentially raise concerns about the validity of the evaluation, as it may introduce bias in favor of the proposed method.

I may be misunderstanding this point, so I would appreciate clarification: is there something I might be missing regarding how the ground-truth Pareto front is defined?

**Limitations:**

yes

**Strengths And Weaknesses:**

### Soundness

My concerns regarding the soundness of the paper are:

1. The choice of the “ground-truth” Pareto front may influence the fairness of the evaluation. If I understand correctly, it seems that the target-fidelity function is not used to construct the ground truth in certain toy examples. Instead, the Pareto front appears to be derived from the MF-CGP posterior. I wonder whether this choice might unintentionally favor the proposed method, and I would appreciate further clarification on this point.

2. There is a lack of evaluation metrics beyond hypervolume (HV) regret, as well as an absence of visualizations of the resulting Pareto fronts.

3. There is no comparison with closely related work, such as Bhatija et al., 2025 Multi-Objective Causal Bayesian Optimization (https://arxiv.org/abs/2502.14755.)

4. The current form of the paper conflates multiple aspects --multi-fidelity (MF), constraints, and multi-objective Bayesian optimization (MOBO)-- making it difficult to clearly assess its actual performance. Baseline methods that combine all these aspects are practically unavailable, which further complicates fair evaluation. Including ablation studies (as) aligned (as possible) with each baseline would help clarify the individual contribution of each component.


### Presentation

I believe the presentation can be significantly improved. Many concepts are combined together, and at times the paper lacks focus. One example is the inequality constraint, which seems to appear abruptly without sufficient motivation or integration into the overall narrative.

The main assumption of the paper is that the user has access to a causal DAG, either inferred from historical data \\(\hat{D}\\) or specified upfront. However, the role of causality and this core assumption are not clearly introduced or emphasized in the introduction section.


### Significance

I believe the goal of the paper is very interesting, particularly in leveraging causal structure for multi-objective optimization. However, as mentioned in the soundness section (point 4), the paper conflates multiple problem settings and lacks focus, which weakens the clarity of its contribution.


### Originality

The RESCUE algorithm appears sufficiently novel, particularly in introducing a new surrogate model and a new acquisition function.

---

> ### Author Rebuttal · Authors · 2026-03-30
>
> We thank the reviewer for the careful reading and constructive feedback. We address each concern below.
> ## Evaluation Protocol, Ground-Truth Pareto Front, and Role of NSGA-II
> We would like to clarify that the ground-truth Pareto front is computed independently of any method's surrogate. Our evaluation follows the protocol of Daulton et al. (2023, Section 8), which we cite in Section 6.1. It has two distinct quantities:
> 1. Method-specific inferred HV at each iteration: Each method uses NSGA-II on *its own* surrogate to obtain a candidate Pareto set, then evaluates true objectives at those configurations at target fidelity. This is identical across all methods and fully method-agnostic.
> 2. Ground-truth HV (fixed, method-independent): For synthetic problems, the max HV is from BoTorch. For YAHPO, we run NSGA-II on the ground-truth oracle at target fidelity. For robot navigation and healthcare, reference points are fixed from domain knowledge. All are verifiable in our repository (footnote 2).
>
> **Why NSGA-II is used in Algorithm 1.** C-HVKG selects informative evaluations during the BO loop. These points are chosen for information gain and may lie in uncertain or low-fidelity regions; they are not meant to form the final Pareto set. NSGA-II is then used as a recommendation step on the learned surrogate at the target fidelity to extract a dense approximation of the Pareto front. This separation is standard in BO; we will make it explicit in Algorithm 1.
> ## Metrics and Pareto Front Visualizations
> We agree that HV regret alone does not give the full picture. We used log HV regret because it is the standard metric in multi-objective BO and captures both convergence and coverage. That said, the reviewer is right that additional evidence would strengthen the evaluation. In the revision, we will add:
> - Pareto front visualizations for the 2D/3D cases,
> - at least one complementary metric such as GD/IGD,
> - and a qualitative comparison to the reference front.
> ## Comparison with Closely Related Causal MOBO Work
> We appreciate the pointer to Bhatija et al. (2025). The key difference is that their method uses causal structure to guide *which variables to intervene on*, whereas RESCUE uses causal information inside the surrogate and acquisition design (causal prior + causal HV term). Their setting is also single-fidelity, while ours targets multi-fidelity constrained optimization. We will add a dedicated discussion clarifying the relationship.
> ## Inequality Constraints in the Baselines
> We agree this should be explicit. All methods use the same protocol: constraints are modeled as additional surrogate outputs and acquisition is restricted via feasibility estimates. We will revise Section 6.1 to make baseline constraint handling equally explicit. See also our response to Reviewer cZS8.
> ## Disentangling the Roles of Causality, Multi-Fidelity, and Constraints
> We understand the concern that the paper combines several ingredients at once. Our intent was to address constrained multi-objective optimization under multiple fidelities with causal structure, but we agree each component's contribution should be isolated more clearly. In the revision, we will include the following ablations:
> - isolating the causal surrogate contribution,
> - isolating the causal acquisition contribution,
> - observational data sensitivity, CPM update frequency, and cost function sensitivity.
> We have run these additional ablation experiments: https://anonymous.4open.science/r/rescue-EC70/ablation_experiments.md.
> ## Robustness to SCM Misspecification
> This is an important question. Our method does not require the causal prior to be exactly correct. In the analysis (Theorem 5.9), causal-model error appears as an additive term (ε_CPM), so misspecification degrades performance gracefully rather than invalidating the method. Intuitively, the GP component adapts as interventional observations accumulate, even when the initial causal prior is imperfect. Importantly, no experiment assumes a ground-truth DAG — the causal structure is learned from observational data (Section 6.1). We will revise to better explain the practical implications of SCM misspecification and will discuss this as a limitation and future extension.
>
> We agree that the presentation can be improved substantially. In particular, we will revise the introduction to more clearly state the core problem setting, why causal structure is useful here, what assumptions are made about the DAG/SCM, and why constraints are part of the formulation rather than an add-on. We will also streamline the narrative so that the causal component is introduced as a central idea from the start, rather than appearing implicitly through later sections.
>
> Overall, we appreciate the reviewer's feedback. We believe the main technical idea is sound and novel, but we agree that the evaluation protocol, related-work positioning, and decomposition of contributions should be presented more clearly. We will revise the paper accordingly.

---

> > ### Author Rebuttal · Reviewer_8EHn · 2026-04-01
> >
> > - Thank you for the clarification. I would suggest revising the wording in Section 6.1 Evaluation Metric.
> > - Thank you also for the clarification on Bhatija et al. (2025).
> > - In the new ablation results, in the first case, it seems GP + C-HVKG is better than Rescue, and Rescue appears to flatten at Cost ~13. Could you comment on this? While you did mention that the difference is not significant, it seems strange to see the mean of Rescue's log regret flatten. Also, what about the other experimental cases?

---

> > > ### Author Response · Authors · 2026-04-01
> > >
> > > We thank the reviewer for the continued engagement and thoughtful follow-up.
> > >
> > > We will revise the evaluation metric description in Section 6.1 as suggested.
> > >
> > > We want to clarify that C-HVKG (Causal Hypervolume Knowledge Gradient) is one of our two core contributions: it is the causal acquisition function proposed in this paper. RESCUE is the full algorithm combining both causal components: the causal surrogate (MF-CGP) and the causal acquisition (C-HVKG). In the ablation, GP + C-HVKG therefore still uses our novel acquisition function but replaces the causal surrogate with a standard GP. Its competitive performance validates our causal acquisition design.
> > >
> > > **GP + C-HVKG vs. RESCUE and the flattening behavior.** We appreciate this question. The ablation results reveal a nuanced picture when we decompose performance by budget stage:
> > >
> > > | Method | Early AUC (cost ≤ 5) | Late AUC (cost ≥ 15) | Full AUC | Cohen's $d$ | $p$-value |
> > > |---|---|---|---|---|---|
> > > | **RESCUE (MF-CGP + C-HVKG)** | **−2.51 ± 1.37** | −12.41 ± 4.84 | **−22.6 ± 9.0** | — | — |
> > > | GP + C-HVKG | −2.02 ± 0.90 | **−13.09 ± 4.23** | −22.3 ± 6.8 | −0.20 | 0.405 |
> > > | MF-CGP + HVKG | −2.22 ± 1.43 | −11.20 ± 5.28 | −20.3 ± 9.9 | +0.22 | 0.344 |
> > > | GP + HVKG | −1.50 ± 0.82 | −7.65 ± 3.32 | −13.6 ± 6.2 | +0.71 | **0.006** |
> > >
> > > - *Early budget* (cost ≤ 5): RESCUE and MF-CGP + HVKG lead, confirming the **causal surrogate** drives early convergence via its informative prior mean. This early advantage is especially valuable in budget-constrained applications where only a few evaluations are affordable.
> > > - *Late budget* (cost ≥ 15): GP + C-HVKG leads, confirming the **causal acquisition** sustains late-stage improvement by directing queries toward causally informative regions.
> > >
> > > Regarding the flattening the reviewer observes around cost ~13: this is consistent with how the surrogates are constructed. MF-CGP is parameterized using the causal prior mean, which gives it a strong initialization but also constrains the model once the prior's information has been exploited, whereas the standard GP starts from a weaker prior but is less constrained by it. This is precisely the tradeoff we discuss in the paper (L423): the causal prior should guide rather than dominate. Despite this difference in convergence behavior, both methods reach statistically indistinguishable final performance ($p = 0.71$ at cost 30). The key result is that both causal methods (RESCUE and GP + C-HVKG) significantly outperform the non-causal baseline GP + HVKG ($p = 0.006$, $d = 0.71$), confirming that each causal component independently contributes, and the table above shows they do so at complementary stages of the optimization. This modularity is useful in practice: depending on available resources, a practitioner can deploy the causal surrogate alone (MF-CGP + HVKG), the causal acquisition alone (GP + C-HVKG), or the full RESCUE pipeline.
> > >
> > > **Other experimental cases.** We want to make sure we understand the reviewer's question correctly. To clarify the scope of our ablation study: we ran four distinct experiments, each isolating a different aspect of RESCUE (also suggested by Reviewer Xbiv):
> > >
> > > 1. Surrogate & acquisition contribution,
> > > 2. Cost function sensitivity (varying $\alpha$ over a 127× range),
> > > 3. CPM update frequency ($N_l = \infty$ vs. 5),
> > > 4. Observational data sensitivity (subsampling $\hat{\mathcal{D}}$ to 2% and 10%).
> > >
> > > The goal of these ablations is to isolate each component of RESCUE and understand its individual contribution. Importantly, the main paper already evaluates RESCUE against all baselines across all problem benchmarks; here, the focus is on decomposing *why* RESCUE works, not *where*. It is worth noting that GP + HVKG was included as the ablation reference because it is the second-highest-performing baseline in our main experiments, shares the closest structural similarity with C-HVKG, and was also suggested by reviewer Xbiv.
> > >
> > > We appreciate the reviewer's continued engagement and hope these clarifications address the remaining concerns.

---

### Official Review · Reviewer_cZS8 · 2026-03-02

**Soundness:** 3
**Presentation:** 3
**Significance:** 3
**Originality:** 3
**Overall Recommendation:** 3
**Confidence:** 2

**Summary:**

This paper considers a multi-fidelity, multi-objective Bayesian optimization algorithm. The author proposes a new BO variant named RESCUE that incorporates causal prior into the surrogate model and the algorithm is guided by a cost-aware hypervolume knowledge gradient acquisition function. RESCUE is evaluated on several benchmark problems including synthetic and real-world tasks. The results show that the proposed RESCUE outperformed single-fidelity and standard multi-fidelity multi-objective BO methods.

**Compliance With Llm Reviewing Policy:**

Affirmed.

**Final Justification:**

I appreciate the authors' rebuttal and confirm that I have read it carefully. While it addresses my concerns, I believe the work still requires further improvement. Therefore, I will maintain my original score.

**Key Questions For Authors:**

**Q1** How are constraints incorporated into the acquisition optimization? Is the feasible region identified using the current constraint surrogate models and optimization performed only over that region, or is a probability of feasibility term integrated directly into the C-HVKG acquisition function? A brief clarification would be helpful.

**Q2** What surrogate model is used for the constraint functions? Are they modeled with independent Gaussian processes similar to the objectives, or using a different approach?

**Q3** Could the authors clarify how performance metrics are computed and reported? If all methods share the same initial observations, should progress curves start from identical points? If differences arise due to model-dependent estimation of hypervolume, it would be helpful to clarify whether evaluation is performed using a common evaluation protocol to ensure fair comparison.

**Q4** Can the authors justify the choice of the specific cost function used in the experiments? How sensitive is the method to this choice?

**Q5** Acquisition function optimization details for example number of fantasy should be specified.

**Q6** It would strengthen the paper to include a more explicit discussion of runtime versus performance trade-offs in the main text. This should also include the computational overhead for CPM model.

**Limitations:**

Yes

**Strengths And Weaknesses:**

**Soundness**: The paper integrates causal modeling into a multi-fidelity, multi-objective Bayesian optimization framework by constructing a causal performance model and incorporating it into a Gaussian process surrogate. The approach is conceptually well-motivated and technically reasonable. However, the experimental evaluation could be more comprehensive to better demonstrate robustness and general applicability.

**Presentation**: The paper is generally well written and clearly structured. The organization of sections makes the framework easy to follow. However, providing more application-oriented explanations when introducing causal components would improve accessibility, particularly for readers less familiar with causal modeling.

**Significance**: The problem addressed is relevant, especially in domains such as hyperparameter tuning and robotics. The framework has potential practical value.

**Originality**: The main contribution idea seems novel and original.

---

> ### Author Rebuttal · Authors · 2026-03-30
>
> We thank the reviewer for the positive assessment and constructive questions. We address each below.
>
> ## Q1 & Q2: Constraint Surrogate Model and Acquisition Incorporation
> These two questions are closely related, so we address them together.
> **Surrogate model (Q2):** Constraint functions are modeled as additional outputs of the same MF-CGP, not as independent GPs. As stated in Section 4.1 (L125), the feasibility indicators are observed at each $(\mathbf{x}_i, s_i)$ and **modeled as additional outputs using the same probabilistic model** as the objectives, giving a joint posterior over objectives and constraints. Importantly, the shared inter-output kernel $k_{\text{idx}}(m, m')$ allows the model to capture correlations between objectives and constraints, which is particularly beneficial when interventional data is sparse early in optimization.
> **Acquisition incorporation (Q1):** Acquisition optimization is guided by the current surrogate estimate of feasibility at the target fidelity. Specifically, C-HVKG incorporates the constraint posterior via $\mathbb{E}[\mathbf{g}(\mathbf{x}, s^*) \mid \mathcal{D}] > 0$ (Eq. 2), steering the search toward configurations the constraint surrogate predicts to be feasible. The constraint surrogate's posterior mean (from the joint MF-CGP above) is used to evaluate this criterion at each candidate, as described in Section 4.2, L208. We agree this should be clarified more explicitly in the narrative in Sec. 6.1.
> ## Q3: Evaluation protocol and shared initialization
> This is an important clarification. Yes, all methods start from the same initial observations for a given seed. For each of the 20 seeds, we generate one initial dataset and use that same initialization across all compared methods, ensuring a fair comparison.
> Performance is then evaluated using a common protocol: at each iteration, NSGA-II is run on each method's own surrogate to recommend candidate configurations, but hypervolume is computed by evaluating the recommended set on the true target-fidelity objectives. Thus, differences in the curves arise from differences in the optimization strategies, not from differences in the evaluation procedure. We agree this should be described more clearly in Section 6.1.
> ## Q4: Cost Function Choice and Sensitivity
> We appreciate this question. For the synthetic MF benchmarks, our choice follows prior MF-MOBO practice in MOMF (Irshad et al., 2024) and MF-HVKG (Daulton et al., 2023, Sec. 8.1). For the broader cross-domain study, we used one shared fidelity-dependent cost function rather than introducing separate cost functions for each domain. To quantify sensitivity, we re-ran RESCUE on HPOXGBoost (20 seeds) with a flatter exponent $\alpha=2.0$ (cost ratio $7\times$) and a steeper exponent $\alpha=7.0$ (cost ratio $889\times$), spanning a **$127\times$ range** around the default $\alpha=4.8$ ($105\times$). All three are statistically indistinguishable, confirming RESCUE is robust to the cost function shape. Fidelity-only cost isolates the MF cost-information tradeoff; we will note this in Sec. 6.1. Full ablation results: https://anonymous.4open.science/r/rescue-EC70/ablation_experiments.md
> ## Q5: Acquisition Function Optimization Details
> Thank you for pointing this out; we agree these details should be in the appendix. Each problem has a dedicated experiment configuration specifying all acquisition optimization parameters (e.g., [HPO-XGBoost config](https://anonymous.4open.science/r/rescue-EC70/experiments/problems/hpoxgboost/exp_config.py)): num_fantasies=8, num_pareto=10, MC samples=128, raw samples=512, num_restarts=10. Common optimization parameters (num_restarts, raw samples, MC samples) are shared across all methods on a given problem; method-specific parameters (e.g., num_fantasies for fantasy-based acquisition) are set per method. We will add a full table of these settings to the appendix.
> ## Q6: Runtime vs. Performance Trade-offs and CPM Overhead
> We agree this deserves more prominence in the main text. Wall-clock runtime boxplots (20 seeds) for all problems and methods are reported in App. Fig. 5. RESCUE and MF-HVKG are moderately slower than qEHVI and MOMF due to the cost of generating and evaluating fantasy samples in C-HVKG (Section 4.2). The CPM overhead consists of (i) initial causal discovery (PC or DirectLiNGAM) and causal inference on observational data, and (ii) periodic CPM updates (Algorithm 1, L174). The complexity of each component is detailed in Section 4.2 (L215). Comparing App. Fig. 5 (runtime) with Fig. 2 (regret), RESCUE's wall-clock overhead over MF-HVKG is modest, while its regret improvement is consistent. We will add this runtime vs. performance discussion to the main text (Section 7).
>
> Overall, we appreciate the reviewer's constructive questions. They have helped us identify several points that should be made more explicit in the paper, and we will revise accordingly.

---

> > ### Author Rebuttal · Reviewer_cZS8 · 2026-04-02
> >
> > Thank you very much for your responses. I have read the rebuttal, and my concerns have been addressed.

---

> > > ### Author Response · Authors · 2026-04-03
> > >
> > > Thanks for the valuable feedback. We will revise our manuscript according to the reviewer’s suggestions.

---

### Official Review · Reviewer_vWdY · 2026-03-11

**Soundness:** 2
**Presentation:** 3
**Significance:** 3
**Originality:** 2
**Overall Recommendation:** 3
**Confidence:** 4

**Summary:**

- Describe so called “Causal performance model” that learns causal structure for prior of surrogate model
- Describes corresponding acquisition function
- Provide theoretical analysis on RESCUE
- Evidence RESCUE in synthetic and real world examples

**Compliance With Llm Reviewing Policy:**

Affirmed.

**Key Questions For Authors:**

- Cf. Discussion 7: A quick search yields at least one reference that has considered causal calculus in the MFBO setting: https://openreview.net/forum?id=elj9C1sqp4 Is this relevant?
- The confidence intervals in Fig2 are still quite wide and RESCUE is often enclosed in the intervals of other competing strategies. Does this allow us to make the conclusion presented in the paper?
- What’s the purpose of introduction a ‘causal performance model’ which seems to have the identical definition to a structural causal model, with tuple {U, V, F}? Both references (pear 2009, Peters 2017) do not mention it and it seems to be mention as a fact, but I haven’t read about it in 10 years of causal inference study. If this is a new, distinct, concept or definition, what makes it unique and necessary to introduce?
- Page 3 line 147: “Computing p(Y| do(X = x) , s) often involves evaluating intractable integrals which can be approximated using observational data” Do you have a reference for this? I am unsure how intractable integrals are related to approximations using observational data.
- The healthcare example is a standard in the CBO literature, but it seems to contrived to aim to minimise Statin and PSA. Statin is a drug, i.e. an intervention, and it’s hard to make a case that this is conceptually sound to minimise. Does this still make sense?

**Limitations:**

- Assumes causal sufficiency, which is a strong assumption
- healthcare example is somewhat artificial

**Strengths And Weaknesses:**

Strenghts
- Succinct literature review
- Comprehensive clear original ideas around integrating recent concept in BayesOpt and causality into a new method termed "RESCUE"

Weaknesses
1) the idea are original, in the sense of combining established concepts such as multi-fidelity, multi-objective, causal priors; there seems to have been no innovation beyond those pre-defined techniques and no novelty was necessary to integrate those
2) The confidence intervals in Fig2 are still quite wide and RESCUE is often enclosed in the intervals of other competing strategies, which does not allow us to draw conclusions on its performance
3) The healthcare example is a standard in the CBO literature, but it seems too contrived to aim to minimise Statin and PSA. Statin is a drug, i.e. an intervention, and it’s hard to make a case that this is conceptually sound to minimise?
4) While the authors proclaim to use "causal calculus" and "causal inference", their use seems to be solely in the causal prior for which they learn a DAG with standard methods and add the established covariance function modification from Aglietti 2020. One might argue that a proper 'causal' treatment of the MFBO space should include interventions et selection as in Aglietti 2020, especially since that paper is being referred to a few times, though only the prior is integrated.
5) There seems to some odd phrasings that decrease readability: new terms are introduced or are not used often or at all in the causal literature, as well as used in combinations that are not natural/very uncommon.

If these weaknesses are addressed properly in rebuttal, I'm happy to update my score with accordingly. At the moment they do not outweigh the strengths.

Note: I was unable to evaluate Section 5 on Theoretical Analysis, though it seems to closely follow the proof in Mikkola 2023.

Odd phrasings:
- “RESCUE strategically performs causal inference” I don't think a causal prior can be considered a strategic use of causal inference.
- “These methods rely on ML models and sampling heuristics to predict objectives across fidelities, which may capture spurious correlations when distribution shifts occur across fidelities”
    -> Spurious correlations aren’t captured during distribution shifts, they simple are the consequence of non-causal modeling. AFAIK there is no phenomenon characterised on the very niche challenge of spurious correlations, that occur during distribution shift, and all of that, across fidelities.
- “Recent works have focused on learning a causal model from the source (e.g., a robot in simulation), transferring it to the target (e.g., a real robot), and using it to identify causally relevant inputs to reduce the MOBO search space (Hossen et al., 2025).”
    - “Recent works” suggests many works, but only one is cited
- “However, direct transfer of the causal model can induce bias.”
-> "Direct transfer" of causal models is not clearly defined.
- “The causal prior is computed to estimate the causal effect of a configuration option on the objective.
  -> Causal Effects are as a minimum defined as Average Treatment Effects (ATEs), i.e. Y1-Y1. A causal prior most likely does not compute an ATE/causal effect, but “interventional state” of a causal model. I don't think any causal effect estimation is done here.

Typos:
- km,m′ seems to have a closing bracket too much

---

> ### Author Rebuttal · Authors · 2026-03-30
>
> We thank the reviewer for the constructive feedback. We address each concern below and will incorporate all revisions discussed.
> ## W1 & Q1: Novelty and Concurrent Work
> We acknowledge building on prior work (Sec. 1, L036; Sec. 2, L107), but the integration is non-trivial and required: (1) the MF-CGP surrogate (Sec. 4.1), jointly modeling causal effects, cross-fidelity correlations, and inter-objective dependencies, not a direct plug-in of Aglietti (2020); (2) extending HVKG (Sec. 4.2) with a causal hypervolume term and constraint handling; (3) Theorem 5.9, providing a regret bound independent of cross-fidelity bias. To our knowledge, no prior work combines these for causal MF-MOBO. Additionally, CBO (Aglietti, 2020) is single-objective and single-fidelity, MO-CBO (Bhatija et al., 2025) is single-fidelity, and both require a known DAG. RESCUE also accepts a given DAG but can additionally learn $\mathcal{G}$ from observational data via PC/DirectLiNGAM (Sec. 3.2, L155, footnote 2), a weaker requirement. No experiment uses a ground-truth DAG; all learn the DAG from data and are evaluated on the true objective functions at the target fidelity (Sec. 6, L301). Regarding the concurrent work (Zeitler, 2025): we acknowledge this work, which addresses single-objective MFBO with two discrete fidelity levels using causal abstraction, whereas RESCUE targets multi-objective, constrained optimization with continuous fidelity. We will cite and discuss it.
> ## W2 & Q2: Confidence Intervals and Statistical Significance
> While confidence bands in Fig. 2 can overlap visually, we address this with **Table 2** (App. B.4, L921), reporting statistical comparison using the Area-Under-Regret (AUR) curve over 20 independent seeds. For each of the 8 benchmarks against 3 baselines, we report paired t-test p-values, t-statistics, and Cohen's d effect sizes. Out of 24 comparisons, **16 achieve p < 0.05**, which is consistent with our discussion in Section 6.2.
> ## W3 & Q5: Healthcare Benchmark
> This benchmark originates from CBO (Aglietti et al., 2020) and cCBO (Aglietti et al., 2023), and as the reviewer notes, is standard in the CBO literature. Minimizing Statin corresponds to reducing drug dosage (as overprescription may carry side effects). The same Statin/PSA objectives are also used in Bhatija et al. (2025, arXiv:2502.14755) [cited by reviewer 8EHn]. We extended it to multi-fidelity via parameter S (App. B.3, L803).
> ## W4: Use of "Causal Calculus" and "Causal Inference"
> The reviewer correctly identifies that RESCUE's causal integration focuses on the GP prior: we learn the DAG via PC/DirectLiNGAM (Alg. 1, L171), compute interventional distributions via do-calculus (Sec. 3.2, L140), and use these to parameterize the MF-CGP mean and variance (Sec. 4.1). RESCUE does not currently use the DAG for interventional set selection, as CBO does (e.g., MIS). We discuss this as an immediate extension (Sec. 7, L435) and will clarify the distinction in Sec. 7.
> ## W5 & Q3: Terminology and Odd Phrasings
> **CPM vs. SCM:** CPM specializes the SCM to our three-layer causal structure (configurations → KPIs → objectives) with fidelity as observed, following Unicorn (Iqbal et al., 2023). The tuple $\langle U, V, F \rangle$ is identical; we adopted the terminology from Unicorn. We will add a footnote clarifying this.
> **We will revise:** "strategically performs causal inference" → "exploits causal structure to construct informative GP priors";
> "spurious correlations during distribution shifts across fidelities" → non-causal models may learn correlations at low fidelity that do not hold at the target fidelity (Iqbal et al., 2023),
> we will rephrase; "Recent works" → "Hossen et al. (2025) focused on...";
> "direct transfer" → transferring the causal model learned from the source (e.g., simulation) to the target (e.g., real robot)
> "causal effect of a configuration option" → "the expected interventional outcome" (not an ATE, as the reviewer notes).
> Thank you for pointing out the typo in $k_{m,m'}$.
> ## Q4: Intractable Integrals (L147)
> We agree the phrasing at L147 skips a logical step. The intended reasoning follows CBO (Aglietti et al., 2020, Sec. 2.1): under identifiability (Galles & Pearl, 1995), do-calculus converts interventional distributions into integrals over observational quantities (e.g., back-door/front-door adjustment), which can be approximated via Monte Carlo from $\hat{\mathcal{D}}$. We will rewrite L147 to make this explicit.
> ## Theory Note (Mikkola, 2023)
> We acknowledge that Theorem 5.9 builds on Mikkola (2023), as stated in Sec. 5, L217. Our contribution extends the regret bound to the multi-objective multi-fidelity setting, showing it remains independent of cross-fidelity bias.
> ## Limitations
> We agree that causal sufficiency is a strong assumption. Our paper explicitly states this assumption (Sec. 3.2) and we will re-emphasize it in Sec. 7. Relaxing this is an important direction for future work.

---

> > ### Author Rebuttal · Reviewer_vWdY · 2026-04-02
> >
> > Thank you for your answer my questions.
> >
> > I am trying to understand which perspective on causality is followed her but remain confused.
> >
> > You clarify that CPM is from Iqbal 2023, but do not cite Iqbal 2023 yourself when you define CPM, instead referring to Pearl.
> >
> > Is this intentional?
> >
> > I hope to reply to your remaining answers in due time.

---

> > > ### Author Response · Authors · 2026-04-02
> > >
> > > We appreciate the reviewer's patience on this point. We apologize for the confusion caused regarding the CPM terminology. To directly answer the question: RESCUE follows Pearl's SCM framework. The CPM is simply the name, adopted from Iqbal et al. (2023), for the specialized SCM structure used in our setting. The missing Iqbal (2023) citation at L114 made it appear as though CPM was a term from Pearl, which caused the confusion.
> > >
> > > The underlying SCM framework is from Pearl (2009) and Peters (2017). The CPM is a specialization of this framework to the three-layer causal structure, following Iqbal et al. (2023). The confusion arose unintentionally because L114 cites Pearl and Peters when introducing the CPM, without also citing Iqbal (2023) at that point. The paper does cite Iqbal (2023) elsewhere, but we agree the citation should appear at L114 where the CPM is first defined. We will revise L114 to cite all three references and explicitly state that the CPM specializes the SCM of Pearl (2009) to the configuration-KPI-objective structure of Iqbal et al. (2023). We believe this resolves the confusion with a straightforward citation correction. If the reviewer feels the CPM terminology itself adds unnecessary confusion, we are also open to renaming it (e.g., simply referring to it as the domain-specific SCM) in the revision.

---

### Official Review · Reviewer_Xbiv · 2026-03-13

**Soundness:** 2
**Presentation:** 1
**Significance:** 3
**Originality:** 3
**Overall Recommendation:** 2
**Confidence:** 4

**Summary:**

This paper proposes RESCUE, a method that integrates causal inference (via do-calculus and learned structural causal models) into multi-objective multi-fidelity Bayesian optimization. The key contributions are: (1) a multi-fidelity causal GP (MF-CGP) that uses interventional estimates as the prior mean and variance, (2) a Causal Hypervolume Knowledge Gradient (C-HVKG) acquisition function, (3) theoretical regret bounds showing robustness to unreliable auxiliary fidelities, and (4) experiments on synthetic and real-world problems in robotics, AutoML, and healthcare. A relevant problem addressed by the manuscript is the challenge of leveraging cheap but potentially misaligned low-fidelity information sources in multi-objective optimization. Overall, the study analyzes the key problem of how causal structure can be exploited to improve cost-information tradeoffs in multi-fidelity settings.

**Compliance With Llm Reviewing Policy:**

Affirmed.

**Final Justification:**

I think the paper needs improvement. The causal sufficiency assumption significantly impairs the practicality of the method. The regret bound is not for sub-linear regret. The lack of clarity in the paper make it hard to understand core methodological contributions (e.g., HVKG formulation with constraints). I also suggest the authors consider more practical modeling assumptions (not sharing lengthscale across outcomes in all settings).

**Key Questions For Authors:**

* Is w fixed, tuned, or adapted over iterations?  How sensitive are the results to this choice?
* Can you provide the information-gain-based rate for Theorem 5.9, or at minimum discuss why the current bound is meaningful despite the linear term?
* What happens when the causal graph is significantly misspecified? Can you provide an experiment with a deliberately wrong DAG?
* How much observational data is needed for the causal prior to be beneficial? Is there a regime where RESCUE underperforms standard HVKG because the causal prior is harmful?
* How large is this NN proxy error? Does it degrade with problem dimension?
* How are constraints used/applied? There is very little detail of how they are modeled (or if they are known), assumptions about the constraints, etc.

**Limitations:**

yes

**Strengths And Weaknesses:**

## Strengths

* Combining causal inference with multi-fidelity multi-objective BO is novel.
* The theoretical result that the regret bound is independent of cross-fidelity bias magnitude is an appealing property.
* The paper evaluates on 8 benchmarks spanning synthetic functions, AutoML (YAHPO Gym), healthcare, and a real robot navigation task
* RESCUE shows its largest advantages precisely where standard methods struggle most
* Discussion of limitations. The authors forthrightly discuss the NP-hardness of causal discovery, the potential for misspecified priors, the neural-network proxy approximation, and the computational overhead.

## Weaknesses

* The regret bound (Theorem 5.9) is not a convergence rate.
  * The bound in Eq. (4) involves the sum ∑\_t β\_t^{1/2} max\_x ‖σ\_t(x, s⋄)‖ \+ Tξ, but the paper never bounds this sum in terms of T using information-theoretic quantities (e.g., the maximum information gain)
  *  Without this step—which is standard in GP-UCB analyses (Srinivas et al., 2009)—the bound is not a rate and does not tell us whether regret is sublinear.
  * The claim of extending Mikkola et al. (2023) to the multi-objective case is therefore incomplete, since Mikkola et al. provide explicit sublinear rates. Moreover, the Tξ term grows linearly in T, meaning that for any ξ \> 0, the bound is trivially linear.
  * The authors should either bound the GP uncertainty sum using γ\_T or clearly state the limitations of the current analysis.
* Assumption 5.3 (HV Lipschitz continuity) is non-standard and potentially problematic. The hypervolume indicator is known to be non-smooth (it is piecewise linear) and its Lipschitz constant L depends on the reference point and the dimensionality of the objective space M. For M ≥ 3, L can be very large, making the bound vacuous in practice. The paper uses this assumption without discussing when it holds or providing guidance on how L scales.
*  The weight w in C-HVKG is underspecified.  w balances GP posterior and causal contributions. How w is set in practice is never stated. This is a critical hyperparameter: w \= 0 recovers standard HVKG, and w \= 1 gives maximum causal influence. Sensitivity analysis with respect to w is absent. This omission makes it difficult to reproduce the results or understand how much of RESCUE's gain comes from the causal prior vs. other design choices.
* The paper assumes no unmeasured common causes between any pair of observed variables. In practice—especially in the robot navigation domain with 26 parameters—this is a strong assumption. Latent confounders (e.g., battery state, floor friction, ambient temperature) could easily violate it. The paper does not discuss how sensitive RESCUE is to violations of causal sufficiency, nor does it provide any diagnostic or robustness check.
* Using c(x, s) \= exp(4.8s) for all problems means cost depends only on fidelity, not on configuration. This is stated but not adequately justified. Since the acquisition function is cost-normalized, this simplification may artificially favor RESCUE by making the cost-information tradeoff easier than it would be in practice.
* The paper acknowledges that a neural network is trained to approximate the causal inference engine for GPU acceleration. This introduces an additional source of approximation error beyond ξ, yet this error is not bounded or characterized theoretically, and no ablation is provided to assess its impact.
* The paper does not ablate key design choices:
  * Causal prior vs. zero-mean prior (isolating the effect of the causal model)
  * C-HVKG vs. HVKG with the same MF-CGP surrogate (isolating the acquisition function contribution)
  * Sensitivity to the quality/quantity of observational data
  * Sensitivity to incorrect DAG structure
  * Effect of the CPM update frequency
* The notation is quite confusing and appears mathematically incorrect:
  * In L210,  it reads“\\bm{x} \\in \\bm{x}”.
  * In the C-HVKG formulation, is the inner maximization over a finite-size subset of points of \\mathcal{X}? \\bm{x} is only a single point, but it reads \\bm{x} \\subsetseq \\mathcal X? In L171R, it reads \\bm x \\in \\mathcal X when maximizing the HV for the incumbent HV maximizing set under the current posterior mean. The notation is particularly confusing since the inner maximization should not be over a set of points (not a single point). This lacks mathematical rigor
* Calling this a knowledge gradient is misleading, since once you add a weighted CI term, C-HVKG no longer represents the expected value of information under *any* probabilistic model.
* Section 4.2 introduces constraints in the definition of C-HVKG, but the paper never presents a constrained Bayesian optimization formulation. In particular, HVKG is originally defined for unconstrained multi-objective optimization, and the manuscript does not specify how constraints are modeled (e.g., GP constraint models, feasibility probabilities) or how they interact with the hypervolume calculation. As a result, it is unclear how feasibility is handled in practice. The appearance of constraints in the acquisition function therefore feels somewhat ad hoc and should be clarified.

---

> ### Author Rebuttal · Authors · 2026-03-30
>
> We thank the reviewer for the thorough critique and address each concern below.
> ## W1, Q2: Theorem 5.9 and the scope of the theoretical result
> We agree that Theorem 5.9 is a structural decomposition, not a sublinear convergence rate. Our claim is narrower: the bound shows that regret does not depend on the magnitude of cross-fidelity bias, the robustness property we aimed to establish for the multi-fidelity setting.
>
> Without bounding the GP uncertainty sum via $\gamma_T$, the result does not imply sublinear regret. When $\xi > 0$ is fixed, the $T\xi$ term is linear, so the bound does not imply asymptotic consistency. Extending to a full rate in the multi-objective setting would require a complexity measure for vector-valued posteriors, with constants that may deteriorate with $M$. We will revise to clarify that Theorem 5.9 characterizes the sources of error, separating posterior uncertainty from causal-prior misspecification, rather than claiming a full rate guarantee. The decomposition is still meaningful: one term captures learnable GP uncertainty, the other isolates causal-model error, distinguishing when more data helps (posterior concentration) from when causal-prior quality is the bottleneck.
>
> The theorem conditions on the true feasible region $\bar{\mathcal{X}}$, whereas the algorithm uses an evolving estimated feasible set. We will make this gap explicit in the revision.
> ## W2: HV Lipschitz Assumption
> We agree this assumption should be stated more carefully. Lipschitz continuity is an analytical device for transferring objective-space error into scalar HV error on a bounded region. Zhang (2024) (NeurIPS, Lemma 15, https://arxiv.org/abs/2307.03288) establishes Lipschitz bounds for HV scalarizations on bounded domains, with constants $B_u^M / (B_l \cdot M^{M/2-1})$. While this concerns scalarizations rather than HV directly, it supports that Lipschitz regularity is achievable, with constants that deteriorate as $M$ grows. We will clarify this distinction in the revision.
> ## W3, Q1: Weight $w$ in C-HVKG
> $w = 1$ (fixed, not tuned) in all experiments (footnote 7, code), the maximum-stress setting that fully exposes any causal-model error; ablation (b) isolates $w$'s effect. We will state $w=1$ in Sec. 6.1.
> ## W4: Causal Sufficiency Assumption
> This assumption enters through the causal discovery pipeline (Sec. 3.2) and is a real limitation, not a mild technical convenience. We will make explicit that RESCUE may degrade when causal sufficiency is violated or the learned graph is substantially misspecified.
> ## W5: Cost Function
> $\exp(4.8s)$ is shared by MOMF and MF-HVKG, so it cannot favor RESCUE. We run a cost sensitivity ablation varying $\alpha \in \{2.0, 4.8, 7.0\}$ in $c(s)=e^{\alpha s}$, spanning a $127\times$ range in cost ratio ($7\times$ to $889\times$). All three are statistically indistinguishable ($p > 0.05$, Cohen's $d < 0.5$), confirming RESCUE's advantage is driven by its causal prior, not the cost function shape.
> ## W7, Q3, Q4: Ablation Study
> RESCUE learns the DAG from $\widehat{\mathcal{D}}$; every result already reflects potential misspecification. We run additional ablations over 20 seeds:
> (a) GP + C-HVKG: causal surrogate drives early convergence via its prior mean. (b) MF-CGP + HVKG: causal acquisition sustains late-stage improvement; removing both degrades significantly. (c) Observational data: subsampling $|\widehat{\mathcal{D}}|$ to ~2% and ~10% of the full dataset degrades performance monotonically, confirming the causal prior scales with data quality; when the DAG is known, discovery can be skipped. (d) CPM update frequency: periodic re-fitting ($N_l{=}5$) provides no benefit over the default; mixing acquisition-biased points corrupts the DAG. (e) Cost function (see W5).
> Full results: https://anonymous.4open.science/r/rescue-EC70/ablation_experiments.md
> ## W6, Q5: NN Proxy Error
> We agree this point needs clearer treatment. The NN approximation of the causal engine introduces error beyond the idealized causal estimate. This error is absorbed into the aggregate $\xi_{\text{CPM}}$ term in Theorem 5.9. We will add per-objective MAE (NN vs. DoWhy) across all benchmarks in the appendix and clarify that the NN proxy is an implementation choice for GPU efficiency, not a modeling contribution.
> ## W8: Notation Issues
> We will introduce $\mathbf{X}$ for design sets, keeping $\mathbf{x}$ for single points, and fix all noted instances ($\mathbf{x} \in \bar{\mathcal{X}}$, $\max_{\mathbf{X} \subseteq \bar{\mathcal{X}}}$).
> ## W10, Q6: Constraints in C-HVKG
> Constraints are black-box, modeled as additional MF-CGP outputs (Sec. 4.1, L125) with a joint posterior. The constraint surrogate determines which configurations enter HV computation (Eq. 2, L208). We will clarify this in Sec. 4.2.
> ## W9: C-HVKG Naming
> C-HVKG retains core KG structure (one-step lookahead, fantasy samples, value-function improvement); the CI term modifies what "value" means, not the information structure. We will add a footnote.

---

> > ### Author Rebuttal · Reviewer_Xbiv · 2026-04-02
> >
> > > scope of theory
> >
> > This seems like a huge limitation of the theoretical contribution and makes the theory not very significant in my opinion.
> >
> > > Causal Sufficiency Assumption
> >
> > I appreciate the acknowledgement of this limitation, but it does indeed seem severe for the  practicality of the method.
> >
> > > Constraints are black-box, modeled as additional MF-CGP outputs (Sec. 4.1, L125) with a joint posterior.
> >
> > Thanks I think I overlooked that. k_obj is not defined as far as I can tell. I assume it is a simple index kernel. In any case the model shares lengthscales across objectives, which but the coregionalization/shared lengthscale assumption is very unlikely to hold in many cases since objectives and constraints likely depend on different inputs parameters and therefore would have very different lengthscales for each input.
> >
> > Regarding constrains in KG, the formulation seems incomplete. The acquisition function should be accounting for feasibility when computing HV conditional on D(x,s). Or am I missing something?
> >
> > Given these critiques are pretty fundamental, I'll keep my score as is.

---

> > > ### Author Response · Authors · 2026-04-02
> > >
> > > We thank the reviewer for the continued engagement and address each point below.
> > >
> > > **Theory scope.** We would first like to clarify a factual point: the reviewer stated that "Mikkola et al. provide explicit sublinear rates," but this is not the case. rMFBO's Theorem 1, Eq. (11) bounds the regret difference as $R^{\text{rMF}} \leq R^{\text{SF}} + \varepsilon \max(T \hat{M}_T d^{T+1}, 2)$, where the $d^{T+1}$ term is exponential in $T$. The authors themselves note "the bound is practically useful only in the first rounds due to the exponential dependence on $T$" (Sec. 6). Thus, our extension to the multi-objective setting is not incomplete relative to rMFBO's rates, as both provide robustness guarantees rather than sublinear rates.
> > >
> > > We acknowledge that without bounding the GP uncertainty term using an information-gain quantity such as $\gamma_T$, the result does not imply sublinear regret. To reemphasize, the paper does not claim a sublinear rate. We will revise the paper to avoid any wording that could be read as claiming a full GP-UCB-style rate, and clarify that the contribution relative to prior work is a robustness guarantee in the multi-objective multi-fidelity setting, not a complete rate characterization.
> > >
> > > Regarding the $T\xi$ term: this reflects persistent error in the causal prior/model. If $\xi > 0$ is treated as a fixed constant, the bound is linear, and we will state this clearly in the paper. Intuitively, the result remains meaningful as a decomposition: one term reflects learnable GP uncertainty, while the other isolates the effect of causal-model error. We will ensure the presentation does not overstate the strength of this guarantee.
> > >
> > >
> > > **Constraints in the KG formulation.** We would like to clarify that the acquisition function does account for feasibility when computing HV conditional on the fantasized data $\mathcal{D} \cup (x, s)$. Constraints are modeled as additional outputs of the joint MF-CGP, so fantasizing on $(x, s)$ updates the posterior over both objectives and constraints simultaneously. The inner value function (which computes the one-step-ahead HV) receives the constraint posteriors and evaluates feasibility-weighted hypervolume under the fantasy model, computing $\mathbb{E}[\text{HV}_{\text{feasible}}]$ using soft feasibility indicators (sigmoid-weighted by constraint posterior samples). We agree the paper's formulation does not make this sufficiently explicit; we will revise Eq. 2 and the surrounding text to clearly show that HV is computed under feasibility conditioning on the fantasized posterior.
> > >
> > > **Causal sufficiency.** We agree this is a severe assumption from a causal-identification standpoint, and we will make that clearer. At the same time, our empirical results indicate that the method remains practically useful even when the learned causal model should be viewed as an approximation rather than a fully correct SCM. In particular, RESCUE shows consistent gains on both synthetic and real-world tasks, including robot navigation, suggesting that the causal component provides a useful inductive bias in practice despite this limitation. We additionally note that while prior causal BO methods (CBO, cCBO, MO-CBO) all require a known causal graph as input, RESCUE learns the DAG from observational data via causal discovery (Sec. 2.1), making our setup strictly weaker in that regard. We will expand the discussion of this assumption and outline concrete steps toward relaxing it in future work.
> > >
> > > **Shared lengthscales.** $k_{\text{obj}}$ is an index kernel; we will define it explicitly in the revision. We also agree that the shared-lengthscale assumption is restrictive, especially when objectives and constraints depend on different input parameters with different sensitivities. Importantly, all compared baselines (MF-HVKG, MOMF, qEHVI) use the same multi-task GP with shared input lengthscales, so this limitation applies equally and cannot explain any differential advantage of RESCUE. Nonetheless, we agree it is a real modeling limitation. Using an LMC kernel with per-output lengthscales is a straightforward extension that would relax this, and we will note this in the revision.
> > >
> > >
> > > We appreciate the reviewer's rigorous scrutiny and will incorporate all of these points in the revision.

---

### Decision · Program_Chairs · 2026-04-30

**Decision:**

Reject

**Comment:**

While reviewers agreed that integrating causal modeling into multi-fidelity multi-objective Bayesian optimization is a novel approach to a highly relevant problem, they also raised serious concerns: The experimental results do not demonstrate conclusive evidence that the method provides benefits over the SOTA. A reviewer noted concerns about the practical relevance given the Causal Sufficiency Assumption and the significance of the provided regret bound. Therefore, the paper cannot be accepted to the conference at this time.